# Deriving and interpreting population size estimates for adolescent and young key populations at higher risk of HIV transmission: Men who have sex with men and females who sell sex

**Lisa Grazina Johnston**[1][⊕]*, **Van Kinh Nguyen**[2][⊕], **Sudha Balakrishnan**[3][‡], **Chibwe Lwamba**[3][⊕], **Aleya Khalifa**[3][‡], **Keith Sabin**[4][⊕]

1 Independent Consultant, UNICEF, New York, New York, United States of America, 2 MRC Centre for Global Infectious Disease Analysis, School of Public Health, Imperial College, London, United Kingdom, 3 UNICEF, New York, New York, United States of America, 4 UNAIDS, Geneva, Switzerland

⊕ These authors contributed equally to this work.
‡ SB and AK also contributed equally to this work.
* lsjohnston.global@gmail.com

**Data Availability Statement:** Some of the data underlying the results presented in the study are

## Abstract

Population sizes of adolescent (15- to 19-years) and young (20 to 24-years) key populations at risk for HIV transmission are essential for developing effective national HIV control strategies. We present new population size estimates of adolescent and young men who have sex with men and females who sell sex from 184 countries in nine UNICEF regions using UNAIDS published population size estimations submitted by national governments to derive 15-24-year-old population proportions based on the size of equivalent adult general populations. Imputed sizes based on regional estimates were used for countries or regions where adult proportion estimates were unavailable. Proportions were apportioned to adolescents and young adults based on age at sexual debut, by adjusting for the cumulative percentage of the sexually active population at each age for sex. Among roughly 69.5 million men who have sex with men, 12 million are under the age of 24 years, of whom 3 million are adolescents. There are an estimated 1.4 million adolescent and 3.7 million young females who sell sex. Roughly four and a half million adolescent men who have sex with men and females who sell sex would benefit from early HIV interventions. These population size estimates suggest there are roughly 17 million adolescent and young men who have sex with men and females who sell sex who need HIV prevention services and social support. These data provide evidence for national and international programs to determine how many adolescent and young key populations need essential health services and are living with HIV and other infections. Age disaggregated population sizes inform epidemic models, which increasingly use age-sex structures and are often used to obtain and allocate resources and human capacity and to plan critical prevention, treatment, and infection control programs.

available from (https://kpatlas.unaids.org/). The rest of the data along with a description of the techniques is available at: https//www.respondentdrivensampling.com or https//www.lisagjohnston.com (see data tab) which has a link to https://github.com/kklot/KPsize.

**Funding:** KN acknowledges funding support from the Bill & Melinda Gates Foundation (OPP1190661) and the MRC Centre for Global Infectious Disease Analysis (reference MR/R015600/1). The funders had no role in study design, data collection and analysis, decision to publish, or preparation of the manuscript.

**Competing interests:** The authors have declared that no competing interests exist.

## Introduction

The world has pledged to end AIDS as a public health threat by 2030. Although there has been notable progress in the past decade to end AIDS, key populations at higher risk of HIV, such as men who have sex with men and females who sell sex, make up about one-third of all new HIV infections—an estimated 450,000. They live with substantially higher risk of HIV transmission compared to the remaining population [1, 2]. A 2020 report by UNAIDS found that the risk of contracting HIV among men who have sex with men is 26 times higher compared to heterosexual men (aged 15 to 49) and for females who sell sex is 30 times higher compared to adult women [1]. Men who have sex with men and females who sell sex are less likely to seek vital HIV and other related health care due to societal stigma and discrimination, increased sexual, physical and emotional violence, and laws and policies that criminalize their behaviors [1, 3]. Among men who have sex with men and females who sell sex, adolescents (15- to 19-year-old) and young people (20- to 24-year-old) typically have less resilience, access and ability to protect themselves from HIV and other harmful health events [3, 4]. However, even though adolescent key populations engage in high-risk behaviors, they face significant legal obstacles to obtaining these essential and lifesaving services because of their ages. In most countries, adolescents are legally restricted from obtaining HIV testing, counselling and treatment without parental consent–this can be a significant structural and policy barrier to adolescents who engage in same sex relationships or sell sex and who may not want to disclose their behaviors to a parent or do not have any legal guardian [4, 5]. They face these obstacles as minors, often without family support. Without major improvements and scale-up in HIV prevention, testing and treatment programs that focus on adolescent and young people's unique circumstances and needs, an additional 379,000 children and adolescents are projected to die of AIDS-related diseases between 2017 and 2030 [6, 7].

The sizes of populations at risk for HIV transmission are an essential component for effective national and international HIV control strategies. Population sizes inform epidemic models, which increasingly use age-sex structures [8] and are used to obtain and allocate adequate resources and human capacity for critical prevention and infection control programs. An understanding of the scope and size of populations affected by HIV is the foundation of an effective HIV response. Adolescent and young key populations require a discrete set of services that will differ from those needed and used by adults. Adolescent and young key populations, like their adult peers need access to commodities such as condoms, lubricants, female-controlled contraceptives, pre-exposure prophylaxes and HIV testing. They will also typically need introduction to sexual and reproductive health, comprehensive sexual education, mental health interventions, and parental and peer support [9]. Community organizations can be critical in delivering these services but they also need to plan and mobilize adequate resources. Furthermore, adolescents and young people require different outreach methods to improve access to HIV and other health services, such as youth-friendly programs and digital platforms [10].

Over the past decade there has been substantial progress in estimating the population sizes of adult men who have sex with men and females who sell sex [11]; however, there are no reliable data on those who are adolescents or young. Population size estimations of adult men who have sex with men and females who sell sex are derived mostly from methods incorporated into Integrated Biological and Behavioral Surveillance (IBBS) surveys using probability-based sampling methods, such as respondent driven sampling (RDS) or time location sampling (TLS) [11]. The population size methods most often used with IBBS are the unique object and service multipliers [12–14], wisdom of the crowds [15], capture/recapture (overlapping of two probability-based surveys), and the successive sampling population size estimation

(SS-PSE) [16, 17]. Other methods used to estimate the size of hidden populations include mapping and enumeration [18–20], PLACE [21], multiple source capture-recapture [22] and network scale up [23–25].

This paper presents new estimates of the population sizes of adolescent and young men who have sex with men and females who sell sex to build a foundation upon which effective action can be taken to address the HIV epidemic. In addition, the most up to date prevalence of HIV is presented to show the magnitude of infection among young key populations.

## Methods

Country population size estimates and HIV prevalence data are based on data submitted between 2014 and 2019 by national governments to UNAIDS through the Global AIDS Monitoring system. Data submitted to UNAIDS are displayed in AIDSInfo (aidsinfo.unaids.org). These data are supplemented in the key population atlas (http://www.aidsinfoonline.org/kpatlas/#/home) which includes other estimates that are published in reports and peer-reviewed journals. HIV prevalence data are mostly based on national HIV biological behavioral surveillance survey findings.

The size estimates submitted to UNAIDS are evaluated regarding national representativeness based on a number of factors including the methodologies used to estimate population sizes, the percentage of the population included in the estimations and the methods used to conduct an extrapolation. Estimates that are deemed "subnational" are extrapolated here to reflect a national estimate. The extrapolation of "subnational" estimates uses the UNAIDS regional median population proportion for all reported size estimates multiplied by the adult population (aged 15–49) of the relevant sex (i.e., males for men who have sex with men and females who sell sex). The regional medians are published in Table 1 in the Spectrum Quickstart guide (https://www.unaids.org/sites/default/files/media_asset/QuickStartGuide_Spectrum_en.pdf). The male and female 15–49 populations are drawn from the World Population Prospects (WPP): 2019 Revision. (https://population.un.org/wpp/). In countries or regions where key population proportion estimates in adults were not available, median proportions of the region were imputed, upon which a spatial smoothing was done. In particular, the logit of the proportion was regressed against a global mean of the population size proportion, a varying mean for each of the UNICEF defined subregions formulated as a random effect (26), and a weighted average of countries that share borders formulated as an Intrinsic Conditional Auto-Regressive term (ICAR). The smoothing was done using integrated nested Laplace approximations (INLA) package (1). Based on the calculated proportion and the national population aged 15–49 obtained from World Population Prospects, PSE for each country aged 15–49 ($PSE_{15:49}$) was obtained. This number was then apportioned to adolescents and young adults based on age at sexual debut, by adjusting for the cumulative percentage of the sexually active population at each age $a$ and sex $s$ ($\rho_{sa}$), using data from the Demographic and Health Surveys (DHS), Multiple Indicator Cluster Surveys (MICS), Health Behaviour in School-aged Children (HBSC), and National Survey of Family Growth (NSFG). The sexual debut rate was estimated separately from the prevalence model. In particular, we used a survival model in which the time to the age at first sex was assumed following a log-logistic distribution. Individuals who had never had sex at the time of the interview were considered "right censored" based on the age at the time of the interview. The log of time since birth to age at first sex was then regressed against a global intercept, a region-specific intercept defined by UNICEF, and a neighboring ICAR structure model. The survey weights were scaled to the sample size and incorporated into the likelihood(1). Once the sexual debut distribution by country and sex were calculated, the PSE for age $a$ and sex $s$ was apportioned as $PSE_{sa} = N_{sa}\,\rho_{sa}\,\frac{PSE_{15:49}}{\sum_a N_{sa}\rho_{sa}}$, where

$N_{sa}$ is the total population size age $a$ obtained from WPP2019 and $s$ is male (for men who have sex with men and transgender women) and female (for females who sell sex) is the total population size age $a$ obtained from WPP2019. The source code for the main steps in the smoothing, the sexual debut rate estimate, and the model constraints specifications for can be found at https://github.com/kklot/KPsize. For the purposes of this paper, men who have sex with men can be defined as having anal sex with a man in the past six months or year, females who sell sex can be defined as having exchanged sex for goods or money in the past six months or year.

## Results

Population size estimations were calculated for men who have sex with men and females who sell sex in 184 countries. Table 1 shows the population size estimates by UNICEF region.

Tables 2 and 3, list the regions, countries, size estimations by age groups and percentage of the key population by age groups, percent of equivalent general population by sex and age group and HIV prevalence for those countries that reported it. The UNICEF regions consist of East Asia and Pacific (EAP), Eastern Europe and Central Asia (EECA), East and South Africa (ESA), Latin America and the Caribbean (LAC), Middle East and North Africa (MENA), North America (NA), South Asia (SA), West and Central Africa (WCA) and West and Central Europe (WE).

Differences in sexual initiation led to different distributions of young men who have sex with men across geographic regions. Globally, an estimated 17% of men who have sex with men were between the ages of 15–24; 4% were adolescents. The averaged proportions of adolescent men who have sex with men (among all men who have sex with men) ranged from 2% in East Asia and the Pacific and eastern Europe and central Asia to 10% in eastern and southern Africa (Fig 1). The average proportion of young (20 to 24 years) men who have sex with men among all estimated men who have sex with men ranged from 9% in eastern Europe and central Asia to 20% in eastern and southern and western and central Africa regions. Among an estimated 69.5 million men who have sex with men aged 15–49, an estimated 12 million are under the age of 24 years, of whom 3 million are adolescents. Of all estimated females who sell sex in their respective regions, adolescent females who sell sex comprised from 2% in East Asia and the Pacific and eastern Europe and central Asia to 11% in West and central Africa and young females who sell sex comprised from 9% in eastern Europe and central Asia to 20% in West and central Africa (Fig 2). Adolescent girls who sell sex were estimated at 1.4 million (note: The United Nations does not recognize "sex workers" under the age of 18 and considers girls under age 18 selling sex as exploited youth. We are currently unable to estimate such

**Table 1. Population sizes of men who have sex with men and females who sell sex, by age and UNICEF region.**

| | Men who have sex with men | | | Females who sell sex | | |
|---|---|---|---|---|---|---|
| | 15–19 | 20–24 | 25–49 | 15–19 | 20–24 | 25–49 |
| East Asia and the Pacific | 439,000 | 1,856,000 | 15,684,000 | 163,000 | 696,000 | 6,072,000 |
| Eastern Europe and central Asia | 106,000 | 451,000 | 4,470,000 | 32,000 | 135,000 | 1,413,000 |
| East and southern Africa | 202,000 | 406,000 | 1,442,000 | 343,000 | 694,000 | 2,535,000 |
| Latin America and the Caribbean | 375,000 | 1,253,000 | 7,303,000 | 137,000 | 453,000 | 2,713,000 |
| Middle East and North Africa | 56,000 | 218,000 | 1,742,000 | 50,000 | 191,000 | 1,438,000 |
| North America | 353,000 | 931,000 | 5,492,000 | 53,000 | 140,000 | 846,000 |
| South Asia | 690,000 | 2,092,000 | 10,930,000 | 249,000 | 753,000 | 4,066,000 |
| West and central Africa | 449,000 | 804,000 | 2,747,000 | 289,000 | 517,000 | 1,787,000 |
| West and central Europe | 322,000 | 993,000 | 7,642,000 | 45,000 | 140,000 | 1,124,000 |

**Table 2. Population size estimations for adolescent and young men who have sex with men, 2019.**

| Region | Country | Size estimate by age groups (counts) | | | | Percent of all men who have sex with men by age groups | | | Percent of equivalent male general population by age group | | | HIV prevalence |
|---|---|---|---|---|---|---|---|---|---|---|---|---|
| | | 15–19 | 20–24 | 15–24 | 25–49 | 15–19 | 20–24 | 25–49 | 15–19 | 20–24 | 25–49 | <25 |
| EAP | China | 236530 | 1054650 | 1291170 | 9744730 | 2.14 | 9.56 | 88.3 | 0.06 | 0.29 | 2.66 | 5.6 (2018) |
| EAP | Indonesia* | 86760 | 286950 | 373710 | 1656730 | 4.27 | 14.13 | 81.59 | 0.12 | 0.39 | 2.22 | 23.8 (2015) |
| EAP | Japan | 27170 | 93610 | 120780 | 790350 | 2.98 | 10.27 | 86.74 | 0.11 | 0.37 | 3.1 | 2.8 (2015) |
| EAP | Philippines* | 16490 | 95380 | 111860 | 714750 | 1.99 | 11.54 | 86.47 | 0.06 | 0.32 | 2.39 | 3.7 (2018) |
| EAP | Thailand* | 14480 | 61110 | 75590 | 442240 | 2.8 | 11.8 | 85.4 | 0.09 | 0.36 | 2.62 | 6.2 (2018) |
| EAP | Australia* | 11520 | 33140 | 44660 | 225190 | 4.27 | 12.28 | 83.45 | 0.19 | 0.55 | 3.75 | 1.2 (2014) |
| EAP | Malaysia | 8490 | 32860 | 41350 | 206030 | 3.43 | 13.28 | 83.28 | 0.09 | 0.35 | 2.18 | 15.5 (2017) |
| EAP | South Korea | 6990 | 37640 | 44630 | 337730 | 1.83 | 9.84 | 88.33 | 0.06 | 0.3 | 2.69 | 4.3 (2011) |
| EAP | Vietnam | 6720 | 48420 | 55150 | 692370 | 0.9 | 6.48 | 92.62 | 0.03 | 0.18 | 2.63 | 10.6 (2018) |
| EAP | Myanmar* | 5340 | 35280 | 40610 | 361090 | 1.33 | 8.78 | 89.89 | 0.04 | 0.24 | 2.47 | 4.2 (2018) |
| EAP | North Korea | 5240 | 22640 | 27890 | 162040 | 2.76 | 11.92 | 85.32 | 0.08 | 0.34 | 2.45 | NA |
| EAP | Cambodia | 3610 | 18500 | 22110 | 120240 | 2.53 | 13 | 84.47 | 0.08 | 0.41 | 2.69 | 0.6 (2015) |
| EAP | Papua New Guinea* | 3400 | 11180 | 14590 | 52750 | 5.05 | 16.61 | 78.34 | 0.14 | 0.46 | 2.17 | NA |
| EAP | New Zealand | 2390 | 7100 | 9490 | 40880 | 4.74 | 14.1 | 81.16 | 0.22 | 0.65 | 3.76 | NA |
| EAP | Laos | 1160 | 6590 | 7750 | 48320 | 2.07 | 11.75 | 86.18 | 0.06 | 0.33 | 2.39 | 2.1 (2017) |
| EAP | Mongolia† | 590 | 2520 | 3110 | 21320 | 2.4 | 10.32 | 87.28 | 0.07 | 0.29 | 2.48 | 0.7 (2017) |
| EAP | Singapore | 500 | 4100 | 4590 | 41410 | 1.08 | 8.9 | 90.02 | 0.03 | 0.27 | 2.69 | 23.8 (2015) |
| EAP | Timor-Leste | 330 | 1520 | 1850 | 7960 | 3.39 | 15.46 | 81.15 | 0.1 | 0.44 | 2.32 | NA |
| EAP | Solomon Islands | 240 | 880 | 1120 | 4130 | 4.61 | 16.78 | 78.61 | 0.14 | 0.51 | 2.39 | NA |
| EAP | Fiji | 230 | 880 | 1110 | 5720 | 3.32 | 12.89 | 83.78 | 0.1 | 0.37 | 2.4 | NA |
| EAP | Brunei | 100 | 400 | 500 | 2850 | 3.07 | 11.93 | 85 | 0.08 | 0.31 | 2.19 | NA |
| EAP | Vanuatu | 100 | 360 | 460 | 1880 | 4.37 | 15.4 | 80.22 | 0.13 | 0.47 | 2.43 | NA |
| EAP | Samoa* | <100 | 270 | 360 | 1180 | 5.9 | 17.24 | 76.86 | 0.18 | 0.53 | 2.38 | 0 (2018) |
| EAP | Tonga | <100 | 160 | 210 | 580 | 6.88 | 19.88 | 73.23 | 0.21 | 0.61 | 2.23 | NA |
| EAP | Micronesia (Federated States of) † | <100 | 160 | 200 | 670 | 5.31 | 17.8 | 76.89 | 0.15 | 0.49 | 2.12 | NA |
| EAP | Kiribati† | <100 | 150 | 200 | 680 | 4.95 | 17.29 | 77.76 | 0.15 | 0.52 | 2.32 | NA |
| EECA | Russia | 41670 | 134360 | 176020 | 1468590 | 2.53 | 8.17 | 89.3 | 0.12 | 0.4 | 4.38 | NA |
| EECA | Turkey | 16560 | 101280 | 117840 | 952720 | 1.55 | 9.46 | 88.99 | 0.07 | 0.45 | 4.25 | NA |
| EECA | Ukraine* | 9800 | 41160 | 50950 | 456430 | 1.93 | 8.11 | 89.96 | 0.1 | 0.4 | 4.46 | 6.7 (2017) |
| EECA | Romania | 7490 | 29270 | 36770 | 273240 | 2.42 | 9.44 | 88.14 | 0.17 | 0.66 | 6.17 | 11.6 (2011) |
| EECA | Uzbekistan | 6880 | 34620 | 41500 | 260180 | 2.28 | 11.47 | 86.24 | 0.08 | 0.38 | 2.84 | 2.9 (2018) |
| EECA | Bulgaria | 3730 | 9920 | 13650 | 99180 | 3.31 | 8.79 | 87.9 | 0.24 | 0.64 | 6.41 | 1.3 (2016) |
| EECA | Serbia | 3020 | 15140 | 18160 | 144070 | 1.86 | 9.33 | 88.81 | 0.15 | 0.74 | 7 | 2.8 (2013) |
| EECA | Kazakhstan | 2820 | 13960 | 16780 | 149320 | 1.7 | 8.4 | 89.9 | 0.06 | 0.31 | 3.31 | 6.7 (2018) |
| EECA | Tajikistan | 2340 | 11680 | 14010 | 72960 | 2.68 | 13.43 | 83.89 | 0.1 | 0.48 | 3 | 1.7 (2017) |
| EECA | Belarus* | 2240 | 8120 | 10360 | 96540 | 2.1 | 7.59 | 90.31 | 0.1 | 0.37 | 4.45 | 4.0 (2013) |
| EECA | Croatia | 1480 | 7630 | 9110 | 66140 | 1.97 | 10.14 | 87.9 | 0.17 | 0.85 | 7.41 | 1.5 (2013) |
| EECA | Turkmenistan | 1280 | 5760 | 7040 | 46200 | 2.4 | 10.82 | 86.78 | 0.08 | 0.36 | 2.9 | NA |
| EECA | Kyrgyzstan† | 1260 | 6540 | 7800 | 50700 | 2.16 | 11.18 | 86.66 | 0.08 | 0.39 | 3.03 | 5.7 (2017) |
| EECA | Bosnia & Herzegovina | 1010 | 6470 | 7480 | 53470 | 1.66 | 10.61 | 87.72 | 0.14 | 0.87 | 7.15 | NA |
| EECA | Azerbaijan* | 1010 | 7160 | 8170 | 85910 | 1.07 | 7.61 | 91.32 | 0.04 | 0.27 | 3.2 | 0.8 (2018) |
| EECA | Albania | 960 | 5840 | 6800 | 48500 | 1.73 | 10.57 | 87.7 | 0.13 | 0.81 | 6.7 | NA |
| EECA | Moldova* | 740 | 3730 | 4470 | 41400 | 1.61 | 8.13 | 90.26 | 0.07 | 0.35 | 3.9 | 7.3 (2017) |
| EECA | Georgia* | 430 | 2800 | 3220 | 32480 | 1.19 | 7.83 | 90.98 | 0.05 | 0.31 | 3.59 | 8.8 (2018) |
| EECA | Armenia* | 370 | 2230 | 2590 | 25530 | 1.3 | 7.91 | 90.78 | 0.05 | 0.32 | 3.67 | 0.6 (2018) |

*(Continued)*

**Table 2.** (Continued)

| Region | Country | Size estimate by age groups (counts) | | | | Percent of all men who have sex with men by age groups | | | Percent of equivalent male general population by age group | | | HIV prevalence |
|---|---|---|---|---|---|---|---|---|---|---|---|---|
| | | 15–19 | 20–24 | 15–24 | 25–49 | 15–19 | 20–24 | 25–49 | 15–19 | 20–24 | 25–49 | <25 |
| EECA | Macedonia* | 350 | 2540 | 2890 | 36480 | 0.9 | 6.45 | 92.65 | 0.07 | 0.48 | 6.96 | 4.4 (2017) |
| EECA | Montenegro | 230 | 1180 | 1410 | 10370 | 1.92 | 10.02 | 88.06 | 0.15 | 0.78 | 6.88 | 3.6 (2014) |
| ESA | Ethiopia | 38370 | 82560 | 120930 | 265910 | 9.92 | 21.34 | 68.74 | 0.13 | 0.28 | 0.89 | NA |
| ESA | Angola | 22480 | 35290 | 57770 | 113080 | 13.16 | 20.66 | 66.19 | 0.29 | 0.46 | 1.48 | 8.2 (2011) |
| ESA | Tanzania | 19480 | 38690 | 58160 | 141070 | 9.78 | 19.42 | 70.81 | 0.13 | 0.26 | 0.96 | 15.4 (2014) |
| ESA | Uganda | 18960 | 33270 | 52220 | 100090 | 12.45 | 21.84 | 65.71 | 0.17 | 0.31 | 0.92 | NA |
| ESA | Kenya | 16750 | 34930 | 51680 | 128320 | 9.31 | 19.41 | 71.29 | 0.12 | 0.24 | 0.89 | 12.2 (2011) |
| ESA | Mozambique | 13840 | 21130 | 34970 | 63190 | 14.1 | 21.53 | 64.37 | 0.19 | 0.28 | 0.85 | NA |
| ESA | Madagascar | 12070 | 23640 | 35700 | 80370 | 10.4 | 20.36 | 69.24 | 0.17 | 0.34 | 1.14 | 9.0 (2014) |
| ESA | South Africa* | 11640 | 30790 | 42420 | 172250 | 5.42 | 14.34 | 80.24 | 0.07 | 0.19 | 1.05 | 0 (2018) |
| ESA | Sudan | 10070 | 27520 | 37590 | 103540 | 7.13 | 19.5 | 73.36 | 0.09 | 0.25 | 0.94 | 0.8 (2015) |
| ESA | Malawi* | 7400 | 13180 | 20580 | 41730 | 11.88 | 21.15 | 66.97 | 0.16 | 0.28 | 0.88 | NA |
| ESA | Zambia† | 7060 | 12290 | 19350 | 39120 | 12.07 | 21.02 | 66.9 | 0.15 | 0.27 | 0.86 | NA |
| ESA | Somalia | 5710 | 11590 | 17310 | 32490 | 11.47 | 23.28 | 65.24 | 0.16 | 0.31 | 0.88 | NA |
| ESA | Zimbabwe† | 5560 | 9590 | 15150 | 30010 | 12.32 | 21.23 | 66.45 | 0.16 | 0.27 | 0.85 | NA |
| ESA | South Sudan | 3830 | 7780 | 11610 | 26370 | 10.09 | 20.48 | 69.44 | 0.14 | 0.28 | 0.94 | NA |
| ESA | Rwanda | 3420 | 8060 | 11470 | 31530 | 7.94 | 18.74 | 73.32 | 0.1 | 0.25 | 0.97 | 2.3 (2016) |
| ESA | Burundi | 1850 | 6600 | 8450 | 31400 | 4.64 | 16.56 | 78.8 | 0.06 | 0.23 | 1.1 | 1.1 (2011) |
| ESA | Eritrea | 1190 | 2110 | 3300 | 8680 | 9.93 | 17.61 | 72.45 | 0.14 | 0.24 | 0.99 | NA |
| ESA | Botswana† | 520 | 1360 | 1880 | 6520 | 6.21 | 16.16 | 77.63 | 0.08 | 0.22 | 1.04 | NA |
| ESA | Namibia | 500 | 1570 | 2070 | 6950 | 5.52 | 17.43 | 77.05 | 0.08 | 0.24 | 1.05 | NA |
| ESA | Lesotho† | 450 | 1260 | 1710 | 5880 | 5.94 | 16.59 | 77.47 | 0.07 | 0.21 | 0.98 | NA |
| ESA | Swaziland† | 360 | 840 | 1200 | 2980 | 8.64 | 20.07 | 71.28 | 0.12 | 0.27 | 0.97 | NA |
| ESA | Mauritius | 250 | 700 | 950 | 4320 | 4.69 | 13.34 | 81.97 | 0.08 | 0.21 | 1.32 | 5.5 (2015) |
| ESA | Djibouti | 230 | 530 | 760 | 2500 | 7.1 | 16.34 | 76.57 | 0.08 | 0.18 | 0.84 | NA |
| ESA | Comoros | 150 | 570 | 720 | 2970 | 3.94 | 15.48 | 80.58 | 0.06 | 0.25 | 1.31 | 0 (2018) |
| ESA | Seychelles | <100 | <100 | <100 | 360 | 3.93 | 11.25 | 84.82 | 0.06 | 0.18 | 1.38 | 0.6 (2013) |
| LAC | Brazil | 114480 | 391500 | 505970 | 2400290 | 3.94 | 13.47 | 82.59 | 0.2 | 0.68 | 4.19 | NA |
| LAC | Mexico* | 55220 | 230170 | 285390 | 1447490 | 3.19 | 13.28 | 83.53 | 0.16 | 0.68 | 4.27 | 11.9 (2013) |
| LAC | Colombia | 34970 | 105540 | 140510 | 550230 | 5.06 | 15.28 | 79.66 | 0.25 | 0.77 | 4.01 | 9.5 (2016) |
| LAC | Argentina | 25420 | 83390 | 108820 | 495760 | 4.2 | 13.79 | 82 | 0.22 | 0.73 | 4.32 | NA |
| LAC | Venezuela | 22340 | 56360 | 78700 | 311450 | 5.73 | 14.44 | 79.83 | 0.31 | 0.79 | 4.37 | NA |
| LAC | Guatemala | 15070 | 44260 | 59320 | 180300 | 6.29 | 18.47 | 75.24 | 0.31 | 0.91 | 3.7 | 6.4 (2017) |
| LAC | Peru | 13830 | 51730 | 65560 | 383990 | 3.08 | 11.51 | 85.42 | 0.16 | 0.59 | 4.4 | 14.5 (2018) |
| LAC | Haiti† | 13630 | 33320 | 46960 | 142780 | 7.19 | 17.56 | 75.25 | 0.45 | 1.1 | 4.69 | 13.3 (2011) |
| LAC | Dominican Republic* | 10340 | 29060 | 39400 | 143320 | 5.66 | 15.9 | 78.44 | 0.36 | 1.01 | 4.98 | NA |
| LAC | Ecuador | 9530 | 35710 | 45240 | 196400 | 3.94 | 14.78 | 81.28 | 0.2 | 0.76 | 4.15 | NA |
| LAC | Bolivia | 8710 | 26560 | 35270 | 125360 | 5.42 | 16.54 | 78.04 | 0.28 | 0.85 | 4.02 | NA |
| LAC | Chile | 8260 | 31280 | 39540 | 218110 | 3.2 | 12.14 | 84.66 | 0.17 | 0.63 | 4.42 | 9.7 (2016) |
| LAC | Cuba* | 8150 | 24270 | 32420 | 158270 | 4.27 | 12.73 | 83 | 0.31 | 0.94 | 6.12 | 0.8 (2018) |
| LAC | Honduras | 8070 | 23930 | 32000 | 101690 | 6.03 | 17.9 | 76.07 | 0.29 | 0.85 | 3.62 | 8.2 (2018) |
| LAC | Nicaragua | 5240 | 14080 | 19320 | 68530 | 5.96 | 16.03 | 78.01 | 0.29 | 0.77 | 3.77 | 6.3 (2016) |
| LAC | El Salvador* | 4340 | 14840 | 19180 | 62930 | 5.28 | 18.08 | 76.64 | 0.27 | 0.92 | 3.89 | 8.9 (2018) |
| LAC | Paraguay* | 4240 | 15510 | 19750 | 79680 | 4.26 | 15.6 | 80.14 | 0.21 | 0.78 | 4.01 | 12.7 (2017) |

*(Continued)*

**Table 2.** (Continued)

| Region | Country | Size estimate by age groups (counts) | | | | Percent of all men who have sex with men by age groups | | | Percent of equivalent male general population by age group | | | HIV prevalence |
|---|---|---|---|---|---|---|---|---|---|---|---|---|
| | | 15–19 | 20–24 | 15–24 | 25–49 | 15–19 | 20–24 | 25–49 | 15–19 | 20–24 | 25–49 | <25 |
| LAC | Costa Rica† | 2800 | 9210 | 12000 | 56140 | 4.1 | 13.51 | 82.39 | 0.21 | 0.68 | 4.15 | NA |
| LAC | Jamaica* | 2690 | 8030 | 10730 | 39110 | 5.41 | 16.12 | 78.47 | 0.35 | 1.03 | 5.01 | 19.2 (2018) |
| LAC | Panama† | 2560 | 8230 | 10790 | 46490 | 4.47 | 14.37 | 81.16 | 0.23 | 0.73 | 4.13 | 5.7 (2018) |
| LAC | Uruguay | 1840 | 6250 | 8090 | 37960 | 4 | 13.57 | 82.44 | 0.22 | 0.74 | 4.49 | 5.5 (2013) |
| LAC | Guyana | 760 | 2330 | 3090 | 9250 | 6.16 | 18.9 | 74.94 | 0.37 | 1.13 | 4.46 | 4.4 (2014) |
| LAC | Trinidad & Tobago | 680 | 2370 | 3060 | 20370 | 2.92 | 10.12 | 86.96 | 0.19 | 0.67 | 5.73 | NA |
| LAC | Suriname | 450 | 1400 | 1850 | 7380 | 4.9 | 15.14 | 79.96 | 0.29 | 0.9 | 4.75 | 13.9 (2018) |
| LAC | Bahamas† | 350 | 990 | 1340 | 5150 | 5.38 | 15.23 | 79.39 | 0.34 | 0.95 | 4.95 | NA |
| LAC | Barbados | 240 | 650 | 890 | 3830 | 5.08 | 13.69 | 81.23 | 0.36 | 0.98 | 5.81 | 11.3 (2014) |
| LAC | Belize | 240 | 870 | 1110 | 4280 | 4.44 | 16.09 | 79.47 | 0.22 | 0.79 | 3.89 | 10.5 (2012) |
| LAC | St. Lucia* | 140 | 450 | 590 | 2480 | 4.51 | 14.6 | 80.89 | 0.28 | 0.91 | 5.04 | NA |
| LAC | St. Vincent & Grenadines | 110 | 300 | 410 | 1500 | 5.62 | 15.78 | 78.6 | 0.37 | 1.04 | 5.17 | NA |
| LAC | Grenada | <100 | 260 | 350 | 1570 | 4.44 | 13.76 | 81.8 | 0.29 | 0.91 | 5.4 | NA |
| LAC | Antigua & Barbuda | <100 | 240 | 320 | 1280 | 5.04 | 14.88 | 80.08 | 0.33 | 0.99 | 5.32 | NA |
| MENA | Iran* | 15530 | 47110 | 62640 | 387800 | 3.45 | 10.46 | 86.09 | 0.07 | 0.2 | 1.66 | NA |
| MENA | Egypt | 11160 | 45320 | 56480 | 277870 | 3.34 | 13.55 | 83.11 | 0.04 | 0.17 | 1.05 | 6.6 (2017) |
| MENA | Morocco* | 6410 | 16940 | 23350 | 90980 | 5.61 | 14.81 | 79.58 | 0.07 | 0.18 | 0.96 | 4.2 (2017) |
| MENA | Algeria | 6390 | 17640 | 24020 | 119210 | 4.46 | 12.31 | 83.23 | 0.06 | 0.16 | 1.05 | 3.3 (2017) |
| MENA | Iraq | 4380 | 25350 | 29730 | 184220 | 2.05 | 11.85 | 86.1 | 0.04 | 0.23 | 1.7 | NA |
| MENA | Yemen | 2250 | 13030 | 15280 | 92180 | 2.09 | 12.13 | 85.78 | 0.03 | 0.16 | 1.14 | 3.1 (2011) |
| MENA | Tunisia | 1640 | 4810 | 6450 | 30580 | 4.44 | 12.98 | 82.58 | 0.05 | 0.16 | 1.02 | 10.6 (2014) |
| MENA | Syria | 1530 | 9950 | 11480 | 85420 | 1.58 | 10.27 | 88.16 | 0.03 | 0.2 | 1.75 | NA |
| MENA | Israel | 1490 | 9120 | 10610 | 88480 | 1.5 | 9.2 | 89.3 | 0.07 | 0.44 | 4.29 | NA |
| MENA | Saudi Arabia | 1220 | 8240 | 9460 | 151360 | 0.76 | 5.12 | 94.12 | 0.01 | 0.07 | 1.23 | NA |
| MENA | Libya | 1070 | 2990 | 4060 | 19040 | 4.62 | 12.96 | 82.41 | 0.05 | 0.15 | 0.96 | NA |
| MENA | Jordan | 960 | 5490 | 6450 | 44880 | 1.87 | 10.7 | 87.44 | 0.03 | 0.19 | 1.58 | NA |
| MENA | Palestinian Territories | 590 | 3410 | 3990 | 22620 | 2.2 | 12.8 | 85 | 0.04 | 0.25 | 1.67 | NA |
| MENA | Lebanon* | 540 | 3490 | 4030 | 31380 | 1.52 | 9.86 | 88.62 | 0.03 | 0.2 | 1.76 | NA |
| MENA | Kuwait | 140 | 810 | 960 | 20910 | 0.65 | 3.72 | 95.63 | 0.01 | 0.05 | 1.35 | NA |
| MENA | United Arab Emirates | 140 | 2000 | 2140 | 46750 | 0.29 | 4.09 | 95.62 | 0 | 0.04 | 0.9 | NA |
| MENA | Bahrain | <100 | 420 | 510 | 10810 | 0.8 | 3.67 | 95.53 | 0.01 | 0.05 | 1.35 | NA |
| MENA | Oman | <100 | 820 | 890 | 23210 | 0.29 | 3.41 | 96.3 | 0 | 0.03 | 0.95 | NA |
| MENA | Qatar | <100 | 780 | 820 | 14670 | 0.26 | 5.01 | 94.74 | 0 | 0.05 | 0.88 | NA |
| NA | United States | 328120 | 845200 | 1173320 | 4900390 | 5.4 | 13.92 | 80.68 | 0.43 | 1.1 | 6.36 | NA |
| NA | Canada | 24850 | 85350 | 110200 | 591200 | 3.54 | 12.17 | 84.29 | 0.29 | 0.99 | 6.85 | 2.1 (2011) |
| SA | India | 442120 | 1503840 | 1945950 | 8332960 | 4.3 | 14.63 | 81.07 | 0.11 | 0.38 | 2.09 | 3.4 (2015) |
| SA | Bangladesh* | 124950 | 204220 | 329170 | 866380 | 10.45 | 17.08 | 72.47 | 0.27 | 0.43 | 1.84 | 0 (2015) |
| SA | Pakistan* | 69520 | 257140 | 326660 | 1279000 | 4.33 | 16.01 | 79.66 | 0.12 | 0.43 | 2.13 | 3.6 (2016) |
| SA | Afghanistan† | 24580 | 58730 | 83310 | 197790 | 8.74 | 20.89 | 70.36 | 0.24 | 0.57 | 1.92 | 0 (2012) |
| SA | Nepal* | 18860 | 44730 | 63590 | 128770 | 9.8 | 23.25 | 66.94 | 0.27 | 0.63 | 1.82 | 5.3 (2017) |
| SA | Sri Lanka* | 10190 | 22240 | 32430 | 115590 | 6.88 | 15.03 | 78.09 | 0.2 | 0.45 | 2.32 | 0 (2016) |
| SA | Bhutan | 230 | 830 | 1060 | 4850 | 3.9 | 14.04 | 82.05 | 0.09 | 0.34 | 1.97 | NA |
| SA | Maldives | <100 | 440 | 490 | 4460 | 0.87 | 8.99 | 90.15 | 0.02 | 0.17 | 1.75 | NA |
| WCA | Nigeria* | 153970 | 301920 | 455890 | 1084870 | 9.99 | 19.6 | 70.41 | 0.3 | 0.6 | 2.14 | 18.6 (2015) |

*(Continued)*

**Table 2.** (Continued)

| Region | Country | Size estimate by age groups (counts) | | | | Percent of all men who have sex with men by age groups | | | Percent of equivalent male general population by age group | | | HIV prevalence |
|---|---|---|---|---|---|---|---|---|---|---|---|---|
| | | 15–19 | 20–24 | 15–24 | 25–49 | 15–19 | 20–24 | 25–49 | 15–19 | 20–24 | 25–49 | <25 |
| WCA | Congo-Kinshasa* | 76330 | 127070 | 203400 | 412190 | 12.4 | 20.64 | 66.96 | 0.37 | 0.61 | 1.98 | 2.1 (2016) |
| WCA | Niger | 30660 | 40750 | 71410 | 107590 | 17.13 | 22.77 | 60.1 | 0.58 | 0.77 | 2.03 | NA |
| WCA | Côte d'Ivoire | 24390 | 40320 | 64710 | 130260 | 12.51 | 20.68 | 66.81 | 0.37 | 0.61 | 1.96 | 9.5 (2015) |
| WCA | Cameroon* | 22710 | 36700 | 59410 | 127420 | 12.16 | 19.64 | 68.2 | 0.34 | 0.55 | 1.9 | 28.8 (2011) |
| WCA | Mali† | 21970 | 31800 | 53770 | 96150 | 14.65 | 21.21 | 64.14 | 0.46 | 0.67 | 2.02 | 10.5 (2015) |
| WCA | Ghana* | 18570 | 42180 | 60750 | 169670 | 8.06 | 18.31 | 73.64 | 0.23 | 0.51 | 2.06 | NA |
| WCA | Chad | 16770 | 25120 | 41880 | 72680 | 14.63 | 21.92 | 63.44 | 0.43 | 0.64 | 1.86 | NA |
| WCA | Burkina Faso | 16080 | 31750 | 47830 | 105320 | 10.5 | 20.73 | 68.77 | 0.31 | 0.62 | 2.05 | 1.2 (2017) |
| WCA | Guinea | 13180 | 21650 | 34830 | 58440 | 14.13 | 23.21 | 62.66 | 0.42 | 0.69 | 1.85 | 11.4 (2017) |
| WCA | Benin | 8930 | 18000 | 26930 | 61850 | 10.06 | 20.27 | 69.67 | 0.3 | 0.6 | 2.07 | 10.2 (2017) |
| WCA | Senegal | 7650 | 22870 | 30530 | 90460 | 6.33 | 18.91 | 74.77 | 0.19 | 0.57 | 2.26 | 19.1 (2018) |
| WCA | Sierra Leone | 7330 | 11530 | 18870 | 39370 | 12.59 | 19.81 | 67.6 | 0.35 | 0.56 | 1.9 | 5.7 (2011) |
| WCA | Togo* | 6060 | 11530 | 17590 | 42680 | 10.06 | 19.13 | 70.81 | 0.29 | 0.55 | 2.03 | 14.6 (2017) |
| WCA | Central African Republic | 5360 | 7720 | 13090 | 20350 | 16.04 | 23.1 | 60.85 | 0.45 | 0.65 | 1.71 | 5.4 (2017) |
| WCA | Liberia* | 5170 | 7440 | 12610 | 24590 | 13.9 | 20.01 | 66.1 | 0.4 | 0.58 | 1.91 | NA |
| WCA | Congo-Brazzaville | 4860 | 7260 | 12120 | 26140 | 12.71 | 18.96 | 68.32 | 0.35 | 0.53 | 1.9 | 32.2 (2018) |
| WCA | Mauritania | 3680 | 7350 | 11030 | 28600 | 9.27 | 18.55 | 72.17 | 0.31 | 0.61 | 2.39 | NA |
| WCA | Gabon | 1450 | 2460 | 3910 | 12130 | 9.02 | 15.34 | 75.65 | 0.24 | 0.41 | 2.02 | NA |
| WCA | Guinea-Bissau* | 1270 | 2760 | 4030 | 10110 | 8.97 | 19.52 | 71.51 | 0.26 | 0.57 | 2.08 | NA |
| WCA | Gambia† | 1210 | 3460 | 4670 | 12940 | 6.89 | 19.63 | 73.48 | 0.21 | 0.59 | 2.22 | 35.5 (2018) |
| WCA | Equatorial Guinea | 770 | 1820 | 2580 | 8520 | 6.89 | 16.34 | 76.77 | 0.16 | 0.39 | 1.82 | NA |
| WCA | Cape Verde | 250 | 620 | 860 | 3660 | 5.48 | 13.61 | 80.91 | 0.15 | 0.38 | 2.24 | 6.6 (2013) |
| WCA | São Tomé & Príncipe | 180 | 330 | 510 | 1220 | 10.34 | 19.27 | 70.39 | 0.33 | 0.61 | 2.23 | 0.8 (2018) |
| WE | Germany | 64460 | 191230 | 255690 | 1300120 | 4.14 | 12.29 | 83.57 | 0.37 | 1.08 | 7.37 | 1.1 (2016) |
| WE | United Kingdom | 60890 | 150900 | 211790 | 953210 | 5.23 | 12.95 | 81.82 | 0.4 | 1 | 6.31 | 1.5 (2015) |
| WE | France | 44970 | 144190 | 189170 | 1001250 | 3.78 | 12.11 | 84.11 | 0.33 | 1.05 | 7.32 | 1.5 (2011) |
| WE | Italy | 38750 | 121290 | 160030 | 967770 | 3.44 | 10.75 | 85.81 | 0.31 | 0.97 | 7.76 | NA |
| WE | Spain | 16370 | 72170 | 88540 | 789060 | 1.87 | 8.22 | 89.91 | 0.16 | 0.7 | 7.61 | 7.2 (2015) |
| WE | Netherlands | 15100 | 42070 | 57160 | 264870 | 4.69 | 13.06 | 82.25 | 0.41 | 1.13 | 7.14 | NA |
| WE | Poland | 10210 | 48070 | 58280 | 544840 | 1.69 | 7.97 | 90.34 | 0.11 | 0.53 | 6.05 | 1.6 (2014) |
| WE | Sweden | 8820 | 21460 | 30280 | 157250 | 4.7 | 11.44 | 83.85 | 0.39 | 0.96 | 7.03 | 1.0 (2013) |
| WE | Belgium | 8400 | 25540 | 33940 | 182700 | 3.88 | 11.79 | 84.33 | 0.33 | 1 | 7.15 | 0.5 (2015) |
| WE | Austria | 7040 | 19820 | 26860 | 140160 | 4.21 | 11.87 | 83.92 | 0.35 | 0.99 | 6.98 | NA |
| WE | Czechia | 6570 | 15810 | 22380 | 158620 | 3.63 | 8.74 | 87.64 | 0.27 | 0.64 | 6.42 | 1.4 (2011) |
| WE | Portugal | 5440 | 19610 | 25060 | 158690 | 2.96 | 10.67 | 86.36 | 0.26 | 0.92 | 7.46 | 2.8 (2011) |
| WE | Norway | 5430 | 13550 | 18980 | 81730 | 5.39 | 13.46 | 81.15 | 0.42 | 1.05 | 6.35 | NA |
| WE | Denmark | 5290 | 15370 | 20660 | 89890 | 4.78 | 13.91 | 81.31 | 0.42 | 1.21 | 7.06 | NA |
| WE | Hungary | 4930 | 15650 | 20580 | 134580 | 3.17 | 10.09 | 86.74 | 0.22 | 0.69 | 5.97 | 4 (2011) |
| WE | Switzerland | 4430 | 17030 | 21460 | 140200 | 2.74 | 10.53 | 86.73 | 0.23 | 0.89 | 7.29 | 3.8 (2013) |
| WE | Finland | 3840 | 11940 | 15780 | 84660 | 3.82 | 11.89 | 84.29 | 0.32 | 1 | 7.08 | NA |
| WE | Ireland* | 2900 | 9360 | 12250 | 72890 | 3.4 | 10.99 | 85.61 | 0.25 | 0.8 | 6.26 | 2.5 (2016) |
| WE | Greece* | 2640 | 14490 | 17120 | 173960 | 1.38 | 7.58 | 91.04 | 0.12 | 0.64 | 7.65 | NA |
| WE | Slovakia | 1390 | 6940 | 8330 | 81710 | 1.54 | 7.71 | 90.75 | 0.1 | 0.52 | 6.12 | NA |
| WE | Slovenia | 970 | 3520 | 4490 | 35050 | 2.44 | 8.91 | 88.64 | 0.21 | 0.77 | 7.67 | NA |

*(Continued)*

**Table 2.** (Continued)

| Region | Country | Size estimate by age groups (counts) | | | | Percent of all men who have sex with men by age groups | | | Percent of equivalent male general population by age group | | | HIV prevalence |
|---|---|---|---|---|---|---|---|---|---|---|---|---|
| | | 15–19 | 20–24 | 15–24 | 25–49 | 15–19 | 20–24 | 25–49 | 15–19 | 20–24 | 25–49 | <25 |
| WE | Lithuania | 890 | 4000 | 4890 | 41200 | 1.94 | 8.67 | 89.39 | 0.16 | 0.7 | 7.23 | 0 (2011) |
| WE | Latvia | 850 | 2320 | 3160 | 28440 | 2.68 | 7.33 | 89.99 | 0.21 | 0.58 | 7.16 | 3.1 (2011) |
| WE | Estonia | 600 | 1810 | 2410 | 20530 | 2.61 | 7.89 | 89.5 | 0.2 | 0.6 | 6.8 | 0 (2018) |
| WE | Cyprus | 320 | 2030 | 2350 | 16350 | 1.71 | 10.84 | 87.45 | 0.1 | 0.64 | 5.17 | NA |
| WE | Luxembourg | 320 | 1270 | 1590 | 10350 | 2.68 | 10.6 | 86.72 | 0.2 | 0.8 | 6.52 | NA |
| WE | Iceland | 290 | 800 | 1090 | 5130 | 4.69 | 12.84 | 82.47 | 0.36 | 0.98 | 6.32 | NA |
| WE | Malta | 220 | 830 | 1040 | 6960 | 2.69 | 10.33 | 86.98 | 0.21 | 0.8 | 6.75 | NA |

*Based on adult data assessed as nationally adequate;

† Based on adult data assessed as nationally inadequate but regionally adequate;

all other countries have either no documented size estimates and/or used inadequate methods.

exploited youth under 18) and there were an estimated 3.7 million 20-24-year-old females engaged in sex work. Limited HIV prevalence data indicates a sizable proportion of adolescent and young boys who have sex with males and females who sell sex are living with HIV.

## Discussion

These population size estimates of adolescent and young men who have sex with men and females who sell sex in 184 countries suggest there are roughly 17 million adolescent and young males who have sex with males and females who sell sex. UNAIDS estimates approximately 10% of new infections among people 15–49 years old occur among young key populations (Unpublished data, personal report, last author). These data can be used to shape and improve the response to the HIV epidemic in these countries. The countries with the highest proportions of men who have sex with men and females who sell sex comprising the equivalent population, (i.e., general male population for men who have sex with men and general female population for females who sell sex) were often found in less populous countries. The highest proportions of adolescent and young men who have sex with men varied widely among countries; the highest proportions of adolescent females who sell sex are found in sub–Saharan Africa, and the highest proportions of young females who sell sex are in Latin America and the Caribbean region.

There are still limited data available on HIV prevalence among adolescent and young key populations. HIV prevalence among men who have sex with men and females who sell sex under the age of 25 exceeded 20% in many reporting countries. UNAIDS estimates that about 70% of HIV infections among males 15–24 occurred among men who have sex with men, transmen and male sex workers and 25% of new HIV infections among adolescent and young females were among females who sell sex or transwomen (Unpublished data, personal report, last author). Surveys that include HIV testing often omit people who are under the age 18 for legal or ethical reasons. Nonetheless, HIV prevalence among young people, under 25 years, indicate that HIV acquisition is appreciable in this age group, warranting attention from HIV prevention and treatment services. Currently, most countries require that a parent accompany a minor to get HIV testing and the few countries that allow a minor to receive HIV testing, require parental consent for positive minors to get care and treatment.

**Table 3. Population size estimations for adolescent and young females who sell sex, 2019.**

| EAP | Country | Size estimate by age groups (counts) | | | | Percent of all females who sell sex by age groups | | | Percent of equivalent female general population by age group | | | HIV prevalence |
|-----|---------|-------|-------|-------|-------|-------|-------|-------|-------|-------|-------|-------|
| | | 15–19 | 20–24 | 15–24 | 25–49 | 15–19 | 20–24 | 25–49 | 15–19 | 20–24 | 25–49 | <25 |
| EAP | China | 78840 | 357520 | 436360 | 3515890 | 1.99 | 9.05 | 88.96 | 0.02 | 0.11 | 1.04 | 0.1 (2018) |
| EAP | Indonesia* | 36000 | 119730 | 155730 | 720970 | 4.11 | 13.66 | 82.24 | 0.05 | 0.17 | 0.99 | 4.1 (2015) |
| EAP | Japan | 10550 | 36380 | 46930 | 310200 | 2.95 | 10.19 | 86.86 | 0.04 | 0.15 | 1.27 | NA |
| EAP | Philippines* | 7460 | 43660 | 51120 | 340520 | 1.91 | 11.15 | 86.95 | 0.03 | 0.15 | 1.18 | 0.7 (2015) |
| EAP | Thailand* | 7300 | 31110 | 38410 | 241160 | 2.61 | 11.13 | 86.26 | 0.04 | 0.18 | 1.42 | 2.8 (2017) |
| EAP | Australia | 3800 | 12550 | 16340 | 59990 | 4.97 | 16.44 | 78.59 | 0.16 | 0.54 | 2.59 | 12.7 (2011) |
| EAP | Malaysia | 3310 | 12840 | 16140 | 78890 | 3.48 | 13.51 | 83.01 | 0.04 | 0.15 | 0.90 | 0 (2017) |
| EAP | South Korea | 2850 | 8300 | 11150 | 58000 | 4.12 | 12.00 | 83.88 | 0.05 | 0.14 | 0.98 | 0 (2013) |
| EAP | Vietnam | 2590 | 13850 | 16440 | 124770 | 1.83 | 9.81 | 88.36 | 0.02 | 0.12 | 1.08 | NA |
| EAP | Myanmar* | 2490 | 18190 | 20680 | 269390 | 0.86 | 6.27 | 92.87 | 0.01 | 0.07 | 1.06 | 1.6 (2017) |
| EAP | North Korea | 2120 | 10680 | 12790 | 74190 | 2.43 | 12.28 | 85.29 | 0.05 | 0.23 | 1.62 | 0.7 (2016) |
| EAP | Cambodia | 2060 | 13780 | 15830 | 148400 | 1.25 | 8.39 | 90.36 | 0.01 | 0.09 | 0.97 | 3.7 (2018) |
| EAP | Papua New Guinea* | 1990 | 8560 | 10550 | 62650 | 2.72 | 11.69 | 85.59 | 0.03 | 0.13 | 0.97 | NA |
| EAP | New Zealand | 590 | 1750 | 2340 | 10910 | 4.48 | 13.18 | 82.34 | 0.05 | 0.16 | 0.99 | NA |
| EAP | Laos | 570 | 3240 | 3810 | 24060 | 2.03 | 11.63 | 86.33 | 0.03 | 0.16 | 1.21 | 1 (2017) |
| EAP | Mongolia† | 220 | 940 | 1160 | 8110 | 2.35 | 10.16 | 87.49 | 0.03 | 0.11 | 0.94 | 0 (2017) |
| EAP | Singapore | 180 | 1280 | 1460 | 14120 | 1.15 | 8.20 | 90.66 | 0.01 | 0.09 | 1.03 | NA |
| EAP | Timor-Leste | 140 | 610 | 750 | 3240 | 3.38 | 15.36 | 81.26 | 0.04 | 0.18 | 0.97 | NA |
| EAP | Solomon Islands | <100 | 330 | 420 | 1700 | 4.35 | 15.58 | 80.07 | 0.05 | 0.20 | 1.00 | 0 (2017) |
| EAP | Fiji | <100 | 360 | 450 | 2330 | 3.29 | 12.97 | 83.73 | 0.04 | 0.16 | 1.03 | 0 (2015) |
| EAP | Brunei | <100 | 150 | 190 | 1040 | 3.24 | 12.01 | 84.75 | 0.03 | 0.13 | 0.88 | NA |
| EAP | Vanuatu | <100 | 130 | 170 | 780 | 3.89 | 14.15 | 81.96 | 0.05 | 0.17 | 1.00 | NA |
| EAP | Samoa* | <100 | <100 | 120 | 410 | 6.10 | 17.14 | 76.76 | 0.07 | 0.20 | 0.91 | 0 (2018) |
| EAP | Tonga | <100 | <100 | <100 | 230 | 6.11 | 17.45 | 76.44 | 0.07 | 0.19 | 0.85 | NA |
| EAP | Micronesia (Federated States of)† | <100 | <100 | <100 | 240 | 5.30 | 17.78 | 76.92 | 0.06 | 0.19 | 0.81 | NA |
| EAP | Kiribati† | <100 | <100 | <100 | 270 | 4.56 | 16.43 | 79.01 | 0.05 | 0.18 | 0.88 | NA |
| EECA | Russia | 13960 | 45230 | 59190 | 525540 | 2.39 | 7.74 | 89.88 | 0.04 | 0.13 | 1.56 | NA |
| EECA | Turkey | 3790 | 23360 | 27160 | 228740 | 1.48 | 9.13 | 89.39 | 0.02 | 0.11 | 1.03 | NA |
| EECA | Ukraine* | 3180 | 13490 | 16670 | 157870 | 1.82 | 7.73 | 90.45 | 0.03 | 0.13 | 1.56 | 1.3 (2017) |
| EECA | Romania | 2950 | 14920 | 17870 | 116590 | 2.20 | 11.10 | 86.71 | 0.03 | 0.17 | 1.29 | 1.7 (2018) |
| EECA | Uzbekistan | 1320 | 5090 | 6410 | 47870 | 2.43 | 9.38 | 88.19 | 0.03 | 0.12 | 1.15 | 1.4 (2011) |
| EECA | Bulgaria | 1240 | 6090 | 7330 | 70140 | 1.60 | 7.86 | 90.54 | 0.03 | 0.13 | 1.55 | 0.6 (2017) |
| EECA | Serbia | 970 | 4920 | 5890 | 32060 | 2.57 | 12.95 | 84.48 | 0.04 | 0.21 | 1.34 | 2.5 (2018) |
| EECA | Kazakhstan* | 750 | 2730 | 3480 | 34150 | 2.00 | 7.25 | 90.75 | 0.04 | 0.13 | 1.60 | 3.8 (2017) |
| EECA | Tajikistan* | 650 | 1710 | 2360 | 17170 | 3.31 | 8.76 | 87.93 | 0.04 | 0.12 | 1.19 | 0 (2016) |
| EECA | Belarus* | 570 | 2580 | 3150 | 21420 | 2.33 | 10.48 | 87.19 | 0.04 | 0.16 | 1.34 | NA |
| EECA | Croatia | 550 | 2860 | 3410 | 23040 | 2.08 | 10.81 | 87.11 | 0.03 | 0.17 | 1.39 | 0.9 (2016) |
| EECA | Turkmenistan | 480 | 3540 | 4020 | 48090 | 0.92 | 6.80 | 92.28 | 0.02 | 0.14 | 1.84 | 4.1 (2018) |
| EECA | Kyrgyzstan | 400 | 2020 | 2430 | 20000 | 1.80 | 9.02 | 89.18 | 0.02 | 0.10 | 1.01 | 0 (2013) |
| EECA | Bosnia & Herzegovina | 240 | 1240 | 1470 | 14190 | 1.52 | 7.89 | 90.59 | 0.02 | 0.12 | 1.36 | 0 (2017) |
| EECA | Azerbaijan* | 210 | 1060 | 1270 | 9420 | 1.92 | 9.95 | 88.13 | 0.02 | 0.12 | 1.10 | NA |
| EECA | Albania | 170 | 1130 | 1310 | 15170 | 1.04 | 6.88 | 92.08 | 0.02 | 0.13 | 1.70 | 0 (2017) |
| EECA | Moldova* | 140 | 870 | 1010 | 7400 | 1.67 | 10.38 | 87.95 | 0.02 | 0.12 | 1.03 | NA |
| EECA | Georgia | 130 | 770 | 900 | 11510 | 1.01 | 6.21 | 92.78 | 0.02 | 0.10 | 1.54 | 0 (2018) |
| EECA | Armenia* | 120 | 770 | 890 | 6220 | 1.65 | 10.88 | 87.46 | 0.02 | 0.12 | 0.94 | NA |

*(Continued)*

**Table 3.** (Continued)

| EAP | Country | Size estimate by age groups (counts) | | | | Percent of all females who sell sex by age groups | | | Percent of equivalent female general population by age group | | | HIV prevalence |
|---|---|---|---|---|---|---|---|---|---|---|---|---|
| | | 15–19 | 20–24 | 15–24 | 25–49 | 15–19 | 20–24 | 25–49 | 15–19 | 20–24 | 25–49 | <25 |
| EECA | Macedonia | <100 | 330 | 380 | 4860 | 0.89 | 6.36 | 92.75 | 0.01 | 0.07 | 0.97 | 0 (2018) |
| EECA | Montenegro | <101 | 150 | 180 | 1420 | 1.79 | 9.47 | 88.74 | 0.02 | 0.10 | 0.98 | 0 (2015) |
| ESA | Ethiopia† | 65470 | 140580 | 206050 | 461120 | 9.81 | 21.07 | 69.12 | 0.22 | 0.48 | 1.56 | NA |
| ESA | Angola | 46940 | 93970 | 140920 | 346500 | 9.63 | 19.28 | 71.09 | 0.32 | 0.64 | 2.37 | NA |
| ESA | Tanzania | 34820 | 63460 | 98270 | 199860 | 11.68 | 21.28 | 67.04 | 0.31 | 0.56 | 1.75 | NA |
| ESA | Uganda | 31910 | 67130 | 99040 | 251960 | 9.09 | 19.13 | 71.78 | 0.22 | 0.46 | 1.74 | NA |
| ESA | Kenya | 26150 | 40330 | 66480 | 127580 | 13.47 | 20.78 | 65.74 | 0.33 | 0.51 | 1.63 | NA |
| ESA | Mozambique | 23900 | 37770 | 61670 | 123840 | 12.88 | 20.36 | 66.76 | 0.30 | 0.48 | 1.57 | 7.2 (2011) |
| ESA | Madagascar | 19180 | 50820 | 70000 | 286820 | 5.38 | 14.24 | 80.38 | 0.12 | 0.31 | 1.76 | NA |
| ESA | South Africa | 16580 | 32740 | 49320 | 111620 | 10.30 | 20.34 | 69.35 | 0.23 | 0.46 | 1.58 | 4.5 (2016) |
| ESA | Sudan | 13440 | 24160 | 37610 | 79390 | 11.49 | 20.65 | 67.86 | 0.27 | 0.49 | 1.62 | NA |
| ESA | Malawi | 13090 | 23140 | 36230 | 75770 | 11.69 | 20.66 | 67.65 | 0.28 | 0.49 | 1.62 | NA |
| ESA | Zambia† | 11690 | 32060 | 43750 | 126040 | 6.89 | 18.88 | 74.23 | 0.11 | 0.29 | 1.14 | 0.4 (2015) |
| ESA | Somalia† | 10060 | 17870 | 27930 | 65790 | 10.74 | 19.07 | 70.19 | 0.25 | 0.45 | 1.65 | NA |
| ESA | Zimbabwe* | 8040 | 16390 | 24430 | 47430 | 11.19 | 22.80 | 66.00 | 0.22 | 0.44 | 1.27 | NA |
| ESA | South Sudan† | 6550 | 13390 | 19940 | 46120 | 9.92 | 20.27 | 69.82 | 0.24 | 0.48 | 1.66 | NA |
| ESA | Rwanda | 6050 | 14320 | 20370 | 59010 | 7.62 | 18.04 | 74.34 | 0.18 | 0.42 | 1.73 | 34 (2016) |
| ESA | Burundi | 3270 | 11780 | 15050 | 57200 | 4.53 | 16.30 | 79.17 | 0.11 | 0.41 | 1.97 | 24.3 (2011) |
| ESA | Eritrea | 1960 | 3560 | 5510 | 14920 | 9.57 | 17.41 | 73.02 | 0.22 | 0.41 | 1.71 | 1.5 (2011) |
| ESA | Botswana† | 920 | 2520 | 3440 | 11160 | 6.27 | 17.27 | 76.46 | 0.16 | 0.44 | 1.94 | NA |
| ESA | Namibia | 820 | 2630 | 3460 | 12000 | 5.33 | 17.02 | 77.65 | 0.12 | 0.38 | 1.75 | NA |
| ESA | Lesotho† | 810 | 2120 | 2940 | 11390 | 5.68 | 14.82 | 79.50 | 0.12 | 0.32 | 1.73 | NA |
| ESA | Swaziland† | 550 | 1220 | 1770 | 5210 | 7.92 | 17.41 | 74.67 | 0.17 | 0.38 | 1.62 | 64.1 (2011) |
| ESA | Mauritius | 340 | 990 | 1330 | 6140 | 4.57 | 13.23 | 82.19 | 0.11 | 0.31 | 1.91 | 5.5 (2015) |
| ESA | Djibouti | 280 | 650 | 930 | 3240 | 6.64 | 15.64 | 77.72 | 0.11 | 0.25 | 1.24 | 21.3 (2018) |
| ESA | Comoros | 200 | 770 | 960 | 4040 | 3.90 | 15.34 | 80.76 | 0.09 | 0.35 | 1.82 | NA |
| ESA | Seychelles† | <100 | <100 | <100 | 460 | 4.31 | 12.22 | 83.47 | 0.10 | 0.29 | 2.01 | 0 (2016) |
| LAC | Brazil | 38090 | 131420 | 169510 | 838900 | 3.78 | 13.03 | 83.19 | 0.07 | 0.23 | 1.46 | 2.1 (2016) |
| LAC | Mexico* | 16520 | 70080 | 86600 | 478580 | 2.92 | 12.40 | 84.68 | 0.05 | 0.20 | 1.36 | 8.1 (2013) |
| LAC | Colombia | 11100 | 33760 | 44860 | 187290 | 4.78 | 14.54 | 80.68 | 0.08 | 0.24 | 1.35 | 4.3 (2013) |
| LAC | Argentina | 9130 | 22860 | 31990 | 101690 | 6.83 | 17.10 | 76.07 | 0.29 | 0.73 | 3.25 | 6.7 (2011) |
| LAC | Venezuela | 8280 | 27300 | 35570 | 169260 | 4.04 | 13.33 | 82.63 | 0.07 | 0.24 | 1.48 | NA |
| LAC | Guatemala | 7960 | 23620 | 31570 | 105020 | 5.83 | 17.29 | 76.89 | 0.16 | 0.47 | 2.11 | NA |
| LAC | Peru | 7200 | 18660 | 25860 | 109670 | 5.31 | 13.77 | 80.92 | 0.10 | 0.26 | 1.50 | NA |
| LAC | Haiti† | 6870 | 19390 | 26260 | 99400 | 5.47 | 15.43 | 79.10 | 0.24 | 0.68 | 3.49 | 2.4 (2012) |
| LAC | Dominican Republic* | 5620 | 16810 | 22430 | 111410 | 4.20 | 12.56 | 83.24 | 0.23 | 0.68 | 4.49 | 0.4 (2018) |
| LAC | Ecuador | 4780 | 19610 | 24390 | 122230 | 3.26 | 13.38 | 83.37 | 0.05 | 0.22 | 1.39 | 1.5 (2017) |
| LAC | Bolivia | 2940 | 11070 | 14010 | 62950 | 3.82 | 14.39 | 81.79 | 0.06 | 0.24 | 1.35 | NA |
| LAC | Chile | 2720 | 8340 | 11060 | 40340 | 5.29 | 16.23 | 78.48 | 0.09 | 0.27 | 1.32 | 0.6 (2012) |
| LAC | Cuba* | 2670 | 10150 | 12830 | 71730 | 3.16 | 12.01 | 84.83 | 0.06 | 0.21 | 1.49 | 0 (2016) |
| LAC | Honduras | 2470 | 7350 | 9820 | 32450 | 5.83 | 17.40 | 76.77 | 0.09 | 0.26 | 1.17 | 2.2 (2018) |
| LAC | Nicaragua | 1790 | 5350 | 7130 | 27870 | 5.11 | 15.27 | 79.62 | 0.23 | 0.68 | 3.54 | 2.6 (2017) |
| LAC | El Salvador* | 1520 | 4100 | 5620 | 22150 | 5.46 | 14.77 | 79.77 | 0.08 | 0.22 | 1.21 | 1.6 (2016) |
| LAC | Paraguay* | 1330 | 4850 | 6170 | 24930 | 4.26 | 15.59 | 80.15 | 0.07 | 0.25 | 1.31 | 0.4 (2017) |

*(Continued)*

**Table 3.** (Continued)

| EAP | Country | Size estimate by age groups (counts) | | | | Percent of all females who sell sex by age groups | | | Percent of equivalent female general population by age group | | | HIV prevalence |
|---|---|---|---|---|---|---|---|---|---|---|---|---|
| | | 15–19 | 20–24 | 15–24 | 25–49 | 15–19 | 20–24 | 25–49 | 15–19 | 20–24 | 25–49 | <25 |
| LAC | Costa Rica† | 1320 | 4700 | 6010 | 23670 | 4.43 | 15.82 | 79.75 | 0.07 | 0.25 | 1.28 | NA |
| LAC | Jamaica | 830 | 2740 | 3570 | 17150 | 4.00 | 13.23 | 82.77 | 0.06 | 0.21 | 1.30 | 0 (2017) |
| LAC | Panama† | 760 | 2460 | 3220 | 14130 | 4.37 | 14.17 | 81.46 | 0.07 | 0.22 | 1.28 | 0.4 (2018) |
| LAC | Uruguay* | 630 | 2150 | 2780 | 13500 | 3.87 | 13.22 | 82.91 | 0.08 | 0.26 | 1.62 | NA |
| LAC | Guyana | 490 | 1400 | 1880 | 5910 | 6.26 | 17.90 | 75.84 | 0.24 | 0.69 | 2.93 | 5.8 (2014) |
| LAC | Trinidad & Tobago | 480 | 1650 | 2130 | 14500 | 2.86 | 9.94 | 87.20 | 0.13 | 0.47 | 4.11 | NA |
| LAC | Suriname | 330 | 1020 | 1350 | 5690 | 4.68 | 14.47 | 80.85 | 0.22 | 0.68 | 3.80 | 9.5 (2018) |
| LAC | Bahamas | 250 | 700 | 940 | 3800 | 5.20 | 14.67 | 80.13 | 0.23 | 0.64 | 3.52 | NA |
| LAC | Barbados | 160 | 450 | 610 | 2870 | 4.71 | 12.94 | 82.35 | 0.25 | 0.68 | 4.31 | NA |
| LAC | Belize | <100 | 310 | 400 | 1780 | 4.32 | 14.17 | 81.52 | 0.19 | 0.61 | 3.52 | NA |
| LAC | St. Lucia* | <100 | 270 | 340 | 1380 | 4.20 | 15.50 | 80.30 | 0.06 | 0.24 | 1.22 | 0 (2012) |
| LAC | St. Vincent & Grenadines | <100 | 200 | 270 | 1020 | 5.36 | 15.43 | 79.20 | 0.24 | 0.70 | 3.62 | NA |
| LAC | Grenada | <100 | 180 | 240 | 1070 | 4.38 | 13.79 | 81.83 | 0.20 | 0.64 | 3.81 | NA |
| LAC | Antigua & Barbuda | <100 | 160 | 220 | 970 | 4.58 | 13.50 | 81.92 | 0.21 | 0.62 | 3.79 | NA |
| MENA | Iran | 12500 | 51060 | 63560 | 323130 | 3.23 | 13.20 | 83.56 | 0.05 | 0.20 | 1.26 | 2.3 (2015) |
| MENA | Egypt | 9650 | 30060 | 39710 | 252920 | 3.30 | 10.27 | 86.43 | 0.04 | 0.13 | 1.09 | 0.7 (2015) |
| MENA | Morocco* | 7500 | 19900 | 27400 | 117220 | 5.19 | 13.76 | 81.06 | 0.08 | 0.21 | 1.21 | 0 (2016) |
| MENA | Algeria | 7400 | 20390 | 27790 | 141870 | 4.36 | 12.02 | 83.62 | 0.07 | 0.18 | 1.28 | 3.5 (2017) |
| MENA | Iraq | 3060 | 17840 | 20900 | 131060 | 2.01 | 11.74 | 86.25 | 0.03 | 0.17 | 1.26 | NA |
| MENA | Yemen | 2360 | 13720 | 16070 | 97500 | 2.07 | 12.08 | 85.85 | 0.03 | 0.17 | 1.24 | NA |
| MENA | Tunisia | 1870 | 5450 | 7320 | 38890 | 4.05 | 11.79 | 84.16 | 0.06 | 0.18 | 1.29 | 0 (2014) |
| MENA | Syria | 1270 | 8000 | 9260 | 103260 | 1.12 | 7.11 | 91.77 | 0.01 | 0.09 | 1.20 | NA |
| MENA | Israel | 1180 | 3360 | 4540 | 21730 | 4.50 | 12.78 | 82.72 | 0.06 | 0.17 | 1.13 | NA |
| MENA | Saudi Arabia | 1010 | 6580 | 7590 | 58890 | 1.52 | 9.89 | 88.59 | 0.02 | 0.14 | 1.22 | NA |
| MENA | Libya | 730 | 4170 | 4900 | 33500 | 1.89 | 10.86 | 87.24 | 0.03 | 0.15 | 1.21 | NA |
| MENA | Jordan | 420 | 2460 | 2880 | 16300 | 2.20 | 12.83 | 84.97 | 0.03 | 0.19 | 1.24 | NA |
| MENA | Palestinian Territories | 370 | 2290 | 2660 | 23190 | 1.44 | 8.86 | 89.70 | 0.02 | 0.11 | 1.15 | NA |
| MENA | Lebanon | 360 | 2390 | 2740 | 23000 | 1.38 | 9.27 | 89.35 | 0.02 | 0.13 | 1.23 | NA |
| MENA | Kuwait | 180 | 1330 | 1510 | 21930 | 0.77 | 5.68 | 93.56 | 0.01 | 0.06 | 1.07 | NA |
| MENA | United Arab Emirates | 140 | 940 | 1090 | 12240 | 1.08 | 7.08 | 91.85 | 0.01 | 0.09 | 1.23 | NA |
| MENA | Bahrain | 120 | 690 | 810 | 11890 | 0.91 | 5.45 | 93.64 | 0.01 | 0.07 | 1.26 | NA |
| MENA | Oman | <100 | 320 | 400 | 4210 | 1.57 | 7.01 | 91.42 | 0.02 | 0.09 | 1.17 | NA |
| MENA | Qatar | <100 | 310 | 360 | 5140 | 0.90 | 5.56 | 93.54 | 0.01 | 0.07 | 1.13 | NA |
| NA | United States | 49410 | 127990 | 177400 | 757000 | 5.29 | 13.70 | 81.01 | 0.07 | 0.17 | 1.00 | NA |
| NA | Canada | 3590 | 12350 | 15950 | 89110 | 3.42 | 11.76 | 84.82 | 0.04 | 0.15 | 1.05 | NA |
| SA | India | 157410 | 534210 | 691620 | 3046660 | 4.21 | 14.29 | 81.50 | 0.04 | 0.15 | 0.84 | 1.2 (2017) |
| SA | Bangladesh* | 45770 | 75610 | 121380 | 338320 | 9.96 | 16.45 | 73.60 | 0.10 | 0.16 | 0.72 | 0.1 (2016) |
| SA | Pakistan* | 25320 | 94840 | 120160 | 484430 | 4.19 | 15.69 | 80.12 | 0.04 | 0.17 | 0.85 | 3.8 (2016) |
| SA | Afghanistan† | 9530 | 22790 | 32320 | 74650 | 8.91 | 21.30 | 69.79 | 0.10 | 0.24 | 0.78 | 0.3 (2012) |
| SA | Nepal* | 6250 | 16140 | 22390 | 66210 | 7.05 | 18.22 | 74.73 | 0.07 | 0.18 | 0.72 | NA |
| SA | Sri Lanka* | 4210 | 9210 | 13420 | 52660 | 6.37 | 13.94 | 79.69 | 0.08 | 0.17 | 0.98 | 0 (2016) |
| SA | Bhutan | 90 | 300 | 390 | 1640 | 4.48 | 14.92 | 80.60 | 0.04 | 0.14 | 0.78 | NA |
| SA | Maldives | 20 | 110 | 130 | 970 | 2.07 | 9.70 | 88.24 | 0.02 | 0.09 | 0.84 | NA |
| WCA | Nigeria | 87130 | 171760 | 258880 | 620220 | 9.91 | 19.54 | 70.55 | 0.18 | 0.35 | 1.26 | 9.8 (2015) |

*(Continued)*

**Table 3.** (Continued)

| EAP | Country | Size estimate by age groups (counts) | | | | Percent of all females who sell sex by age groups | | | Percent of equivalent female general population by age group | | | HIV prevalence |
|---|---|---|---|---|---|---|---|---|---|---|---|---|
| | | 15–19 | 20–24 | 15–24 | 25–49 | 15–19 | 20–24 | 25–49 | 15–19 | 20–24 | 25–49 | <25 |
| WCA | Congo-Kinshasa* | 70640 | 118460 | 189100 | 389800 | 12.20 | 20.46 | 67.33 | 0.34 | 0.57 | 1.87 | 4.5 (2012) |
| WCA | Niger | 19620 | 26000 | 45620 | 74410 | 16.35 | 21.66 | 61.99 | 0.37 | 0.49 | 1.40 | 13.9 (2015) |
| WCA | Côte d'Ivoire | 14020 | 23370 | 37390 | 74570 | 12.52 | 20.87 | 66.60 | 0.21 | 0.35 | 1.12 | 2.4 (2014) |
| WCA | Cameroon | 13740 | 22290 | 36020 | 77840 | 12.06 | 19.57 | 68.36 | 0.21 | 0.33 | 1.16 | 27.5 (2012) |
| WCA | Mali† | 12570 | 18350 | 30920 | 57120 | 14.28 | 20.84 | 64.88 | 0.27 | 0.39 | 1.21 | NA |
| WCA | Ghana | 11760 | 26680 | 38440 | 108290 | 8.01 | 18.19 | 73.80 | 0.15 | 0.34 | 1.37 | 3.4 (2016) |
| WCA | Chad | 10340 | 15460 | 25800 | 44840 | 14.64 | 21.88 | 63.48 | 0.27 | 0.40 | 1.15 | 19.6 (2011) |
| WCA | Burkina Faso | 9130 | 18140 | 27270 | 62570 | 10.16 | 20.19 | 69.64 | 0.18 | 0.36 | 1.23 | 3.6 (2017) |
| WCA | Guinea | 7510 | 12290 | 19800 | 38180 | 12.95 | 21.20 | 65.86 | 0.22 | 0.37 | 1.14 | 10.7 (2017) |
| WCA | Benin | 5100 | 10320 | 15420 | 36610 | 9.80 | 19.84 | 70.36 | 0.17 | 0.35 | 1.23 | 2.9 (2017) |
| WCA | Senegal | 4320 | 6800 | 11110 | 22720 | 12.76 | 20.09 | 67.15 | 0.21 | 0.33 | 1.12 | NA |
| WCA | Sierra Leone | 4260 | 12920 | 17180 | 57300 | 5.72 | 17.34 | 76.94 | 0.10 | 0.30 | 1.35 | 3.3 (2016) |
| WCA | Togo | 3540 | 6730 | 10270 | 25270 | 9.95 | 18.94 | 71.12 | 0.17 | 0.32 | 1.20 | 5.3 (2017) |
| WCA | Central African Republic | 3380 | 4870 | 8250 | 12970 | 15.94 | 22.95 | 61.11 | 0.28 | 0.41 | 1.09 | 14.6 (2014) |
| WCA | Liberia | 3370 | 6740 | 10110 | 26410 | 9.22 | 18.45 | 72.32 | 0.29 | 0.58 | 2.26 | 0 (2014) |
| WCA | Congo-Brazzaville | 3050 | 4570 | 7620 | 16560 | 12.62 | 18.90 | 68.48 | 0.22 | 0.33 | 1.21 | 3.5 (2018) |
| WCA | Mauritania | 2910 | 4240 | 7150 | 14200 | 13.64 | 19.85 | 66.51 | 0.23 | 0.34 | 1.12 | NA |
| WCA | Gabon | 890 | 1520 | 2410 | 6950 | 9.55 | 16.24 | 74.22 | 0.16 | 0.27 | 1.23 | NA |
| WCA | Guinea-Bissau | 730 | 1610 | 2340 | 6250 | 8.48 | 18.74 | 72.78 | 0.14 | 0.32 | 1.23 | 22.2 (2011) |
| WCA | Gambia | 680 | 1980 | 2660 | 7740 | 6.56 | 19.03 | 74.41 | 0.11 | 0.33 | 1.28 | 8.3 (2018) |
| WCA | Equatorial Guinea | 550 | 990 | 1540 | 3760 | 10.40 | 18.70 | 70.91 | 0.17 | 0.31 | 1.17 | NA |
| WCA | Cape Verde | 140 | 340 | 480 | 1870 | 5.95 | 14.40 | 79.66 | 0.09 | 0.22 | 1.23 | 3.9 (2013) |
| WCA | São Tomé & Príncipe | 100 | 180 | 270 | 650 | 10.28 | 19.12 | 70.60 | 0.18 | 0.33 | 1.21 | NA |
| WE | Germany | 8920 | 22220 | 31140 | 145100 | 5.06 | 12.61 | 82.33 | 0.06 | 0.15 | 0.97 | NA |
| WE | United Kingdom | 8670 | 25600 | 34270 | 181490 | 4.02 | 11.87 | 84.12 | 0.05 | 0.15 | 1.09 | 0 (2013) |
| WE | France | 6180 | 20160 | 26340 | 147870 | 3.55 | 11.57 | 84.88 | 0.04 | 0.15 | 1.07 | NA |
| WE | Italy | 5190 | 16070 | 21260 | 136220 | 3.30 | 10.20 | 86.50 | 0.04 | 0.13 | 1.12 | NA |
| WE | Spain | 2190 | 9720 | 11910 | 109060 | 1.81 | 8.04 | 90.15 | 0.02 | 0.10 | 1.08 | 2.2 (2015) |
| WE | Netherlands | 1990 | 5570 | 7550 | 36090 | 4.55 | 12.75 | 82.70 | 0.06 | 0.15 | 1.00 | NA |
| WE | Poland | 1860 | 8840 | 10700 | 97560 | 1.72 | 8.17 | 90.12 | 0.02 | 0.10 | 1.13 | NA |
| WE | Sweden | 1200 | 2870 | 4070 | 21480 | 4.68 | 11.24 | 84.08 | 0.06 | 0.13 | 1.00 | NA |
| WE | Belgium | 1140 | 2750 | 3900 | 27530 | 3.63 | 8.77 | 87.60 | 0.05 | 0.12 | 1.17 | 0.1 (2013) |
| WE | Austria | 1130 | 3430 | 4560 | 25240 | 3.79 | 11.51 | 84.71 | 0.05 | 0.14 | 1.02 | 1.3 (2015) |
| WE | Czechia | 940 | 2640 | 3590 | 19520 | 4.09 | 11.44 | 84.48 | 0.05 | 0.14 | 1.01 | NA |
| WE | Portugal | 850 | 2730 | 3580 | 24070 | 3.09 | 9.86 | 87.05 | 0.04 | 0.12 | 1.10 | NA |
| WE | Norway | 750 | 1830 | 2580 | 11050 | 5.50 | 13.43 | 81.07 | 0.06 | 0.15 | 0.91 | NA |
| WE | Denmark | 730 | 2750 | 3480 | 23690 | 2.70 | 10.12 | 87.19 | 0.03 | 0.12 | 1.07 | 9.1 (2011) |
| WE | Hungary | 680 | 1960 | 2640 | 11840 | 4.71 | 13.53 | 81.75 | 0.05 | 0.16 | 0.95 | NA |
| WE | Switzerland | 600 | 2300 | 2900 | 19300 | 2.68 | 10.36 | 86.96 | 0.03 | 0.12 | 1.02 | NA |
| WE | Finland | 550 | 1700 | 2240 | 12040 | 3.83 | 11.88 | 84.29 | 0.05 | 0.15 | 1.06 | NA |
| WE | Ireland | 410 | 1350 | 1760 | 11000 | 3.22 | 10.54 | 86.24 | 0.04 | 0.11 | 0.94 | NA |
| WE | Greece | 370 | 2050 | 2420 | 24460 | 1.38 | 7.64 | 90.98 | 0.02 | 0.09 | 1.13 | NA |
| WE | Slovakia | 240 | 1180 | 1410 | 13910 | 1.54 | 7.69 | 90.77 | 0.02 | 0.09 | 1.09 | NA |
| WE | Slovenia | 140 | 640 | 780 | 6650 | 1.88 | 8.61 | 89.51 | 0.03 | 0.12 | 1.22 | NA |

(*Continued*)

**Table 3.** (Continued)

| EAP | Country | Size estimate by age groups (counts) | | | | Percent of all females who sell sex by age groups | | | Percent of equivalent female general population by age group | | | HIV prevalence |
|---|---|---|---|---|---|---|---|---|---|---|---|---|
| | | 15–19 | 20–24 | 15–24 | 25–49 | 15–19 | 20–24 | 25–49 | 15–19 | 20–24 | 25–49 | <25 |
| WE | Lithuania | 140 | 360 | 500 | 4690 | 2.67 | 6.94 | 90.39 | 0.04 | 0.09 | 1.21 | NA |
| WE | Latvia | 130 | 480 | 610 | 4650 | 2.51 | 9.04 | 88.45 | 0.03 | 0.11 | 1.11 | NA |
| WE | Estonia | 100 | 290 | 390 | 3170 | 2.67 | 8.25 | 89.08 | 0.03 | 0.10 | 1.13 | NA |
| WE | Cyprus | <100 | 340 | 390 | 3020 | 1.64 | 9.90 | 88.46 | 0.02 | 0.11 | 1.00 | NA |
| WE | Luxembourg | <100 | 170 | 210 | 1370 | 2.67 | 10.45 | 86.88 | 0.03 | 0.11 | 0.91 | NA |
| WE | Iceland | <100 | 110 | 150 | 710 | 4.75 | 12.84 | 82.41 | 0.05 | 0.14 | 0.91 | NA |
| WE | Malta | <100 | 120 | 150 | 990 | 2.76 | 10.47 | 86.77 | 0.03 | 0.12 | 1.02 | NA |

*Based on adult data assessed as nationally adequate;

† Based on adult data assessed as nationally inadequate but regionally adequate;

all other countries have either no documented size estimates and/or used inadequate methods.

The size estimates presented here provide national programmes with data to plan the scale of youth-friendly services. These data also provide evidence for the need to increase flexibility with age limits for HIV and sexual/reproductive health services and treatment, especially for the most vulnerable adolescent populations. This publication offers an initial step in producing usable estimates of the size of adolescent and young vulnerable populations; however, several inconsistencies and limitations in these findings warrant caution in their use. WHO, UNAIDS and GFATM jointly assessed size estimates available by the end of 2019, only 41 countries for males who have sex with males and 19 countries for females who sell sex were deemed to have "nationally adequate" estimates, meaning that they are empirically derived and/or reflect geographic coverage of >50% of the population [1]. Given the importance of size estimates among adolescent and young key populations, as well as adult key populations, more attention

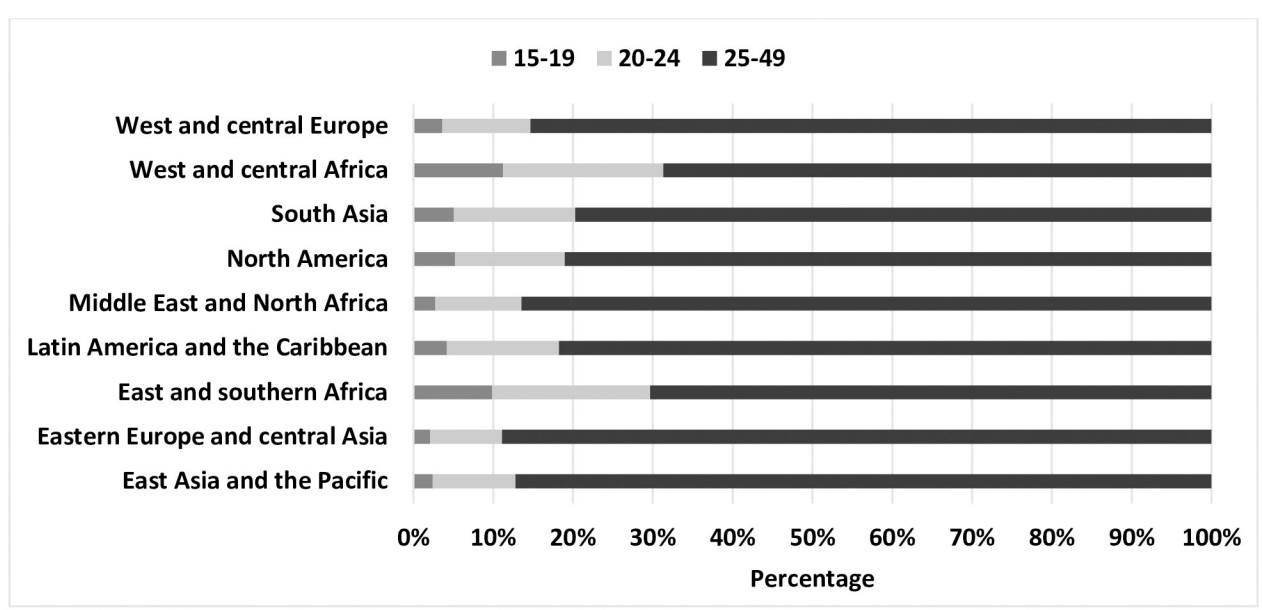

**Fig 1. Distribution of population estimates of men who have sex with men by age group and UNICEF region.**

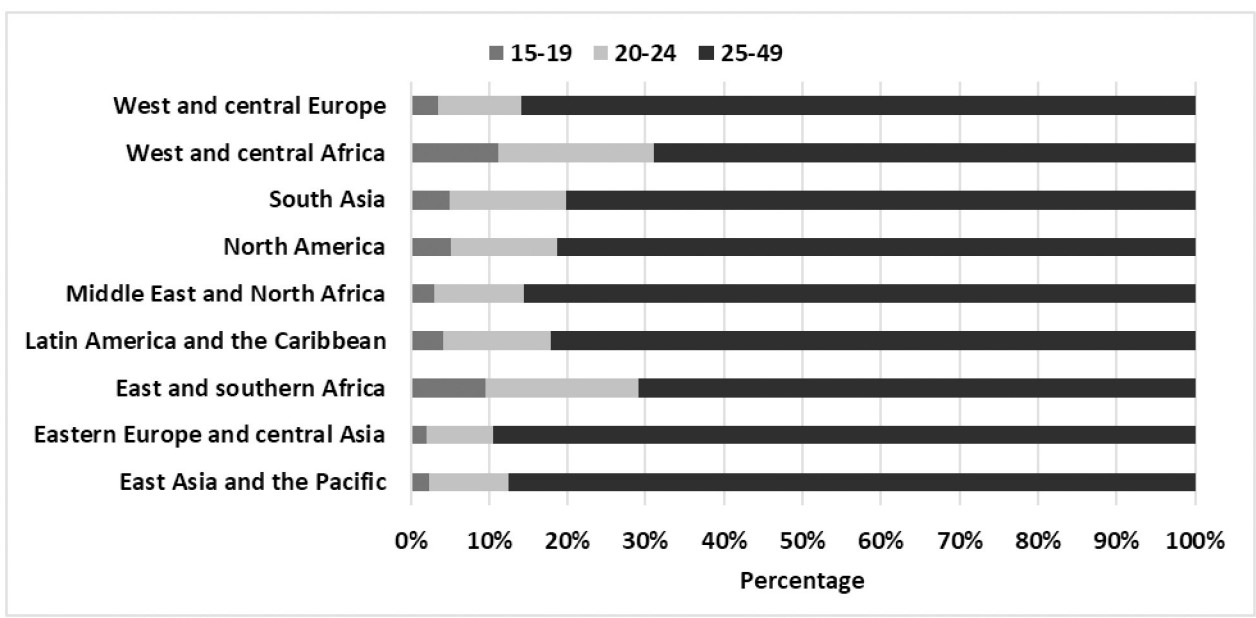

**Fig 2. Distribution of population size estimates of female sex workers by age group and UNICEF.**

is needed for countries to collect the data, formulate their estimates, and clearly describe their methods.

Our estimates are probably non-differentially biased across the breadth of countries but may over- or underestimate in any given country, especially for adolescents since data were adjusted based on the average age at sexual debut of general population adolescents. However, if the bias is directional, it is probably toward underestimation, as sexual debut is likely under-reported. Men who have sex with men and females who sell sex may initiate sex at different ages from the rest of their age cohort. Indeed, we were not able to estimate the population sizes of adolescent and young people who inject drugs due to the lack of data on age of first injection. For countries and regions where surveys on age of first sex were not available, we extrapolated the estimate based on the global and regional trends using only the neighboring structure of the countries and thus did not capture local variations in the age-at-initiation. Future studies could include covariates that might be relevant to culture norms, such as economic status and religion, to improve the extrapolated estimates of the age-at-initiation of sex. Some estimates are likely to be biased by poor implementation of methods, levels of stigma and discrimination in different contexts, and government pressure to minimize the mere existence of these populations [26]. The year in which the original population size estimates were calculated may not correspond with the year (2019) of the general population data. However, reported size estimations were conducted within the past five years, thereby reducing the impact of population fluctuations on the final men who have sex with men and females who sell sex size estimates. The most appropriate estimate of the proportion of adolescents and young key populations would be to use the proportions sampled directly from the survey source, assuming a probability-based sampling method was used. However, many countries do not report these age breakdowns (i.e., 15 to 19, 20 to 24) and many surveys do not sample men who have sex with men or female who sell sex under the age of 18. Furthermore, age-specific adjustments to the proportion of the general population that belongs to each population group would be possible if data were available from men who have sex with men and females who sell sex surveys on the age at sexual debut (or drug injection initiation for young people

who inject drugs). We recommend that countries: 1) include programming for adolescents and young men who have sex with men and females who sell sex in their national HIV response; 2) present disaggregated age in surveys, and 3) calculate population size estimates findings by 15 to 19 years and 20 to 24 years[5, 27]. Such surveys and studies should also collect and publish data on age-at-sexual debut. Ideally, data from surveys of key populations, much like the demographic and health survey data, should be made available for others to conduct further analyses.

Finally, a uniform eligibility criterion is needed. Size estimates vary depending on if the definition of men who have sex with men as "ever having anal sex," "having anal sex in the past one month" or "ever engaged in same sex sexual activity [28]." For the purposes of HIV IBBS surveys, there are recommendations to define men who have sex with men as having anal sex with a man and females who sell sex as having anal or vaginal sex with in the past six [29, 30]. Additionally, transgender women, given their unique risk, stigma and health needs, should be surveyed and counted apart from men who have sex with men. These differences combined with different years or sub-regions of data collection make interpretation challenging.

## Conclusions

Despite the limitations, these findings provide a foundation on which to improve national HIV responses and the general health needs of highly vulnerable young people. Additionally, high quality anthropological and sociological research, as well as deeper secondary analysis from biological behavioral surveillance surveys of these populations, can further enrich understanding and provision of needed services. In addition, these findings show the need for adolescents to have access to age-appropriate health care, including HIV testing, care, and treatment [28]. Adolescent key populations face policy and legal barriers related to age of consent, with third party authorization requirements, which prevent access to health services related to HIV and other sexually transmitted infections and harm reduction. Essential HIV services for adolescent and young key populations will increase the chances that they can protect themselves and receive interventions before they contract HIV. Given the substantial numbers of adolescents practicing high risk behaviors and at risk for HIV exposure, services must find creative ways to engage adolescents and young people from the community. These possibly lifesaving services can be planned and funded based on the estimates presented in this paper, to provide essential, support to an estimated 4 million adolescents and 17 million young key populations around the world. We strongly encourage countries without adequate or any size estimations of these populations to produce them.

## Author Contributions

**Conceptualization:** Lisa Grazina Johnston, Sudha Balakrishnan, Keith Sabin.

**Formal analysis:** Lisa Grazina Johnston, Van Kinh Nguyen, Chibwe Lwamba, Keith Sabin.

**Funding acquisition:** Sudha Balakrishnan.

**Methodology:** Lisa Grazina Johnston, Keith Sabin.

**Writing – original draft:** Lisa Grazina Johnston, Keith Sabin.

**Writing – review & editing:** Lisa Grazina Johnston, Sudha Balakrishnan, Chibwe Lwamba, Aleya Khalifa, Keith Sabin.

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
