## [Decision Letter · Decision Letter 0]

12 May 2022

PONE-D-21-18575Deriving and interpreting population size estimates for adolescent and young men who have sex with men and female sex workersPLOS ONE

Dear Dr. Johnston,

Thank you for submitting your manuscript to PLOS ONE. After careful consideration, we feel that it has merit but does not fully meet PLOS ONE’s publication criteria as it currently stands. Therefore, we invite you to submit a revised version of the manuscript that addresses the points raised during the review process.

Reviewer 2 had some concerns about reliable sources and censoring. I think that you need to write something about this in your response but I am not sure that you can easily address this in the paper as it seems to me an intractable problem. You just need to do what you can.

We look forward to receiving your revised manuscript.

Kind regards,

Andrew R. Dalby, PhD

Academic Editor

PLOS ONE

**Journal Requirements:**

2. Please include your tables as part of your main manuscript and remove the individual files. Please note that supplementary tables (should remain/ be uploaded) as separate ""supporting information"" files.

“KN acknowledges funding support from the Bill & Melinda Gates Foundation (OPP1190661) and the MRC Centre for Global Infectious Disease Analysis (reference MR/R015600/1), jointly funded by the UK Medical Research Council MRC) and the UK Foreign, Commonwealth & Development Office (FCDO), under the MRC/FCDO Concordat agreement and is also part of the EDCTP2 programme supported by the European Union.”

“KN acknowledges funding support from the Bill & Melinda Gates Foundation (OPP1190661) and the MRC Centre for Global Infectious Disease Analysis (reference MR/R015600/1), jointly funded by the UK Medical Research Council MRC) and the UK Foreign, Commonwealth & Development Office (FCDO), under the MRC/FCDO Concordat agreement and is also part of the EDCTP2 programme supported by the European Union. The funders had no role in study design, data collection and analysis, decision to publish, or preparation of the manuscript.”

7. Please amend your list of authors on the manuscript to ensure that each author is linked to an affiliation. Authors’ affiliations should reflect the institution where the work was done (if authors moved subsequently, you can also list the new affiliation stating “current affiliation:….” as necessary).

**Reviewers' comments:**

Reviewer's Responses to Questions

**Comments to the Author**

1. Is the manuscript technically sound, and do the data support the conclusions?

Reviewer #1: Yes

Reviewer #2: Yes

2. Has the statistical analysis been performed appropriately and rigorously? 

Reviewer #1: Yes

Reviewer #2: Yes

3. Have the authors made all data underlying the findings in their manuscript fully available?

Reviewer #1: Yes

Reviewer #2: No

4. Is the manuscript presented in an intelligible fashion and written in standard English?

Reviewer #1: Yes

Reviewer #2: Yes

5. Review Comments to the Author

Reviewer #1: This is a well written and interesting paper that provies sample size estimates for adolescent sex workers in differnet countries. While the accuracy and precision of these estimates are difficult to pin down (both due to various methods used to estimate population sizes and the reliance on underlying data sources), the paper makes interesting contributions by providing cross-country comparisons which could be informative for international collaborative efforts undertaken by international and national public health agencies focused on adolescent sex workers. Despite the limitations (which the authors recognize), I think this is a nice paper.

Reviewer #2: I think the authors did their best efforts to summarize all available data from different sources.

There is a major problem that of course cannot be fully averted by any means.

Notwithstanding, I would like to propose they should be candid on such major challenges, doing their best to incorporate exogenous (besides endogenous) adjustments.

In case this is not possible, re. countries where information is far from free, one should highlight “information to be double-checked”, as explained in detail below. This is a modicum of transparency we need to do in a context of major armed conflicts, collapse of democracy in several countries, ample dissemination of fake news, as well as censorship, worldwide.

The emergence of extreme right and left governments combined with the effects of deeply-entrenched prejudices tend to make some of national estimates nothing but a piece of fiction or a bad version of a fairy tale.

This can be observed in the recent papers showing that “non-existent” COVID epidemics in some African countries have been, at least partially, secondary to misinformation (available at: https://www.bmj.com/company/newsroom/impact-of-covid-19-in-africa-vastly-underestimated-warn-researchers/).

Local governments did NOT provide any help to fix such errors. But at least in the available papers (e.g. the ones from Zambia), censorship has NOT been imposed on accurate information.

In places where misinformation is disseminated in tandem with censorship, no reliable information can be properly obtained and disseminated.

Worst, misinformation backfires: why should international organizations and donors provide vaccines to contexts where epidemics did not take place? Misinformation is a terrible asset for those who want to disseminate false information about unscientific myths, such as a putative “natural immunity” of a given society or country (e.g. the bizarre information from Belarus: https://news.sky.com/story/coronavirus-belarus-president-who-claimed-vodka-could-ward-off-covid-19-says-he-survived-virus-on-his-feet-12038414).

My team and I have experienced such a situation for years. The findings from our population-based survey on the use of substance in Brazil did not match the expectations of the government and have been censored for years (https://www.fairplanet.org/editors-pick/what-the-censorship-of-a-research-now-released-says-about-brazils-deepening-war-on-drugs/).

Similar actions have affected several areas of science (e.g. https://news.mongabay.com/2021/04/intimidation-of-brazils-enviro-scientists-academics-officials-on-upswing/).

The final report was finally cleared, but incorporating a joint statement reached by an agreement between our institution and the government (the full report is available at: https://www.arca.fiocruz.br/bitstream/icict/34614/2/III%20LNUD_ENGLISH.pdf).

The statement allowing peer-reviewed publications eventuating from the original report (there are several peer-reviewed publications and papers currently “in press” despite the substantial delay) made very clear this was/is a “provisional agreement” (The original text is available in Brazilian Portuguese but can be easily understood using a standard translating device and it´s available as follows:

https://www.arca.fiocruz.br/bitstream/icict/34614/12/Nota%20Conjunta%20%c3%a0%20Imprensa.pdf)

Not to make a worldwide problem a personal issue, I would like to cite here a former initiative of a large team of researchers who did their best analyze HIV/AIDS among gay and men who have sex with other men in a large group of South Asian and Middle-East countries. The answer from their respective governments is that there was not a single case of AIDS among this population cause this population does NOT exist!

Of course, there is no way to adjust or carry out any imputation of data about categories that do not even exist!

There is no magic solution for problems such as the ones described above, but authors can and should define strata/rankings of countries where freedom of speech does or does not exist.

Unfortunately, triangulation does not help. The same countries that do not provide reliable data impose strong censorship on peer-reviewed publications. So, one would be cross-comparing non-available data (or even absent conceptual categories) with non-existent papers or papers published under harsh censorship.

I think that, unfortunately, with the collapse of several democracies, worldwide (see, for instance: https://www.amazon.com/When-Democracies-Collapse-Non-Democratic-Democratization/dp/0367888572/ref=sr_1_3?crid=2I4J3IPIM6PJ3&keywords=collapse+of+democracy&qid=1652123086&s=books&sprefix=collapse+of+democracy%2Cstripbooks-intl-ship%2C191&sr=1-3

https://www.amazon.com/Twilight-Democracy-Seductive-Lure-Authoritarianism/dp/1984899503/ref=pd_sbs_sccl_2_1/132-4376952-7835144?pd_rd_w=GgMmk&pf_rd_p=3676f086-9496-4fd7-8490-77cf7f43f846&pf_rd_r=QTP5GB6ZCP9CJ5P23XVQ&pd_rd_r=f960d4e3-9841-466f-935b-ca0b04eb0ea7&pd_rd_wg=vIrLN&pd_rd_i=1984899503&psc=10

… the classic idea of pooling data on sensitive items from diverse societies without any further input from external sources does not longer make sense.

I´m by no means a nihilist thinker, nor one who does not believe world data are useful and key.

I think they must be double-checked against the reliability of sources instead of taken at their face value.

There are several ways to do it:

One is to cross-compare data with the degree of freedom of speech and thinking in different societies and political systems.

There are reliable rankings regularly updated by international agencies addressing such issues from different perspectives.

For instance:

https://worldpopulationreview.com/country-rankings/countries-with-freedom-of-speech

Of course, there is no statistical tool to handle data belonging to conceptual categories that do not exist (how one could provide a reliable estimate on unicorns?). Of course, there are no unicorns besides those from creative books (https://www.amazon.com/Natural-History-Unicorns-Chris-Lavers/dp/0060874147/ref=sr_1_1?crid=3J4POZ347FMN0&keywords=the+natural+history+of+unicorns&qid=1652124098&s=books&sprefix=the+natural+history+of+unicorns%2Cstripbooks-intl-ship%2C216&sr=1-1), whereas key populations do exist everywhere.

Key populations are NOT unicorns, but “unicorns” are a valuable “commodity” for a non-democratic society.

There is no way to extract data from governments who say that in the society “S” gay and other men who make sex with men do not exist cause this would an “abomination” or a” violation of sacred laws” (verbatim)… or that people who use substances deserve capital punishment and that they may exist here and there “as a failure of systems aiming to eliminate this ‘scourge’” (verbatim; I´m just using the words made public by different governments).

There are very interesting analyses about the pronounced influence of funding on the reliability and transparence of clinical science.

See, for instance:

https://pubmed.ncbi.nlm.nih.gov/19596837/

https://pubmed.ncbi.nlm.nih.gov/19596838/

https://pubmed.ncbi.nlm.nih.gov/18616022/

https://pubmed.ncbi.nlm.nih.gov/29401492/

https://pubmed.ncbi.nlm.nih.gov/31497290/

From my point of view, agencies and researchers should apply a similar reasoning to country-based information.

There is no doubt industry and partisan lobbyists may bias scientific findings (see, for instance: https://www.amazon.com/Merchants-Doubt-Handful-Scientists-Obscured/dp/1608193942/ref=sr_1_1?crid=1R8LNEJDL7T1A&keywords=merchants+of+doubt&qid=1652124771&s=books&sprefix=merchant%2Cstripbooks-intl-ship%2C233&sr=1-1)

Non-democratic societies should not be spared of critical scrutiny. The Discussion must be candid about it. NOT in the sense of criticizing A or B, but putting sensitive data on brackets.

Transparent information about the absence of free access to data could be easily shared with the potential readers using international rankings (see, for instance, the excellent reports by the UN agencies such as those available at: https://www.ohchr.org/en/instruments-and-mechanisms). The authors do not need to express their own points of view. The simple dissemination of available information would be very helpful.

Of course, is not our duty (or possible action) “to mend the world”, but to be critical about the information we disseminate. Unfortunately, we live in dark times (there is an ongoing war, but it is not called a “war”!).

There are several lessons to get from Hannah Arendt (e.g. https://www.amazon.com/Men-Dark-Times-Hannah-Arendt/dp/0156588900/ref=sr_1_1?crid=ZU0IKCW5ZQBW&keywords=men+in+dark+times&qid=1652275547&s=books&sprefix=men+in+dark+times%2Cstripbooks-intl-ship%2C214&sr=1-1).

They are not pleasant, but they say the truth, the plain truth.

6. PLOS authors have the option to publish the peer review history of their article (what does this mean?). If published, this will include your full peer review and any attached files.

Reviewer #1: No

Reviewer #2: **Yes: **Francisco Inacio Bastos

---

## [Author Response · Author response to Decision Letter 0]

26 May 2022

Reviewers' comments:

Reviewer's Responses to Questions

Comments to the Author

1. Is the manuscript technically sound, and do the data support the conclusions?

Reviewer #1: Yes

Reviewer #2: Yes

2. Has the statistical analysis been performed appropriately and rigorously?

Reviewer #1: Yes

Reviewer #2: Yes

3. Have the authors made all data underlying the findings in their manuscript fully available?

The PLOS Data policy requires authors to make all data underlying the findings described in their manuscript fully available without restriction, with rare exception (please refer to the Data Availability Statement in the manuscript PDF file). The data should be provided as part of the manuscript or its supporting information or deposited to a public repository. For example, in addition to summary statistics, the data points behind means, medians and variance measures should be available. If there are restrictions on publicly sharing data—e.g. participant privacy or use of data from a third party—those must be specified.

Reviewer #1: Yes

Reviewer #2: No

We have included a statement about why some data are not available. 

4. Is the manuscript presented in an intelligible fashion and written in standard English?

Reviewer #1: Yes

Reviewer #2: Yes

5. Review Comments to the Author

Reviewer #1: This is a well written and interesting paper that provides sample size estimates for adolescent sex workers in different countries. While the accuracy and precision of these estimates are difficult to pin down (both due to various methods used to estimate population sizes and the reliance on underlying data sources), the paper makes interesting contributions by providing cross-country comparisons which could be informative for international collaborative efforts undertaken by international and national public health agencies focused on adolescent sex workers. Despite the limitations (which the authors recognize), I think this is a nice paper.

Thank you for your comments. 

Reviewer #2: I think the authors did their best efforts to summarize all available data from different sources.

There is a major problem that of course cannot be fully averted. I would like to propose they be candid on such major challenges, doing their best to incorporate exogenous (besides endogenous) adjustments.

In case this is not possible, re. countries where information is far from free, one should highlight “information to be double-checked”, as explained in detail below. This is a modicum of transparency we need to do in a context of major armed conflicts, collapse of democracy in several countries, ample dissemination of fake news, as well as censorship, worldwide.

The emergence of extreme right and left governments combined with the effects of deeply-entrenched prejudices tend to make some of national estimates nothing but a piece of fiction or a bad version of a fairy tale.

This can be observed in the recent papers showing that “non-existent” COVID epidemics in some African countries have been, at least partially, secondary to misinformation (available at: https://www.bmj.com/company/newsroom/impact-of-covid-19-in-africa-vastly-underestimated-warn-researchers/).

Local governments did NOT provide any help to fix such errors. But at least in the available papers (e.g. the ones from Zambia), censorship has NOT been imposed on accurate information.

In places where misinformation is disseminated in tandem with censorship, no reliable information can be properly obtained and disseminated.

Worst, misinformation backfires: why should international organizations and donors provide vaccines to contexts where epidemics did not take place? Misinformation is a terrible asset for those who want to disseminate false information about unscientific myths, such as a putative “natural immunity” of a given society or country (e.g. the bizarre information from Belarus: https://news.sky.com/story/coronavirus-belarus-president-who-claimed-vodka-could-ward-off-covid-19-says-he-survived-virus-on-his-feet-12038414).

My team and I have experienced such a situation for years. The findings from our population-based survey on the use of substance in Brazil did not match the expectations of the government and have been censored for years (https://www.fairplanet.org/editors-pick/what-the-censorship-of-a-research-now-released-says-about-brazils-deepening-war-on-drugs/).

Similar actions have affected several areas of science (e.g. https://news.mongabay.com/2021/04/intimidation-of-brazils-enviro-scientists-academics-officials-on-upswing/).

The final report was finally cleared, but incorporating a joint statement reached by an agreement between our institution and the government (the full report is available at: https://www.arca.fiocruz.br/bitstream/icict/34614/2/III%20LNUD_ENGLISH.pdf).

The statement allowing peer-reviewed publications eventuating from the original report (there are several peer-reviewed publications and papers currently “in press” despite the substantial delay) made very clear this was/is a “provisional agreement” (The original text is available in Brazilian Portuguese but can be easily understood using a standard translating device and it´s available as follows:

https://www.arca.fiocruz.br/bitstream/icict/34614/12/Nota%20Conjunta%20%c3%a0%20Imprensa.pdf)

Not to make a worldwide problem a personal issue, I would like to cite here a former initiative of a large team of researchers who did their best analyze HIV/AIDS among gay and men who have sex with other men in a large group of South Asian and Middle-East countries. The answer from their respective governments is that there was not a single case of AIDS among this population cause this population does NOT exist!

Of course, there is no way to adjust or carry out any imputation of data about categories that do not even exist! There is no magic solution for problems such as the ones described above, but authors can and should define strata/rankings of countries where freedom of speech does or does not exist.

Unfortunately, triangulation does not help. The same countries that do not provide reliable data impose strong censorship on peer-reviewed publications. So, one would be cross-comparing non-available data (or even absent conceptual categories) with non-existent papers or papers published under harsh censorship.

I think that, unfortunately, with the collapse of several democracies, worldwide (see, for instance: https://www.amazon.com/When-Democracies-Collapse-Non-Democratic-Democratization/dp/0367888572/ref=sr_1_3?crid=2I4J3IPIM6PJ3&keywords=collapse+of+democracy&qid=1652123086&s=books&sprefix=collapse+of+democracy%2Cstripbooks-intl-ship%2C191&sr=1-3

https://www.amazon.com/Twilight-Democracy-Seductive-Lure-Authoritarianism/dp/1984899503/ref=pd_sbs_sccl_2_1/132-4376952-7835144?pd_rd_w=GgMmk&pf_rd_p=3676f086-9496-4fd7-8490-77cf7f43f846&pf_rd_r=QTP5GB6ZCP9CJ5P23XVQ&pd_rd_r=f960d4e3-9841-466f-935b-ca0b04eb0ea7&pd_rd_wg=vIrLN&pd_rd_i=1984899503&psc=10

… the classic idea of pooling data on sensitive items from diverse societies without any further input from external sources does not longer make sense.

I´m by no means a nihilist thinker, nor one who does not believe world data are useful and key.

I think they must be double-checked against the reliability of sources instead of taken at their face value.

There are several ways to do it:

One is to cross-compare data with the degree of freedom of speech and thinking in different societies and political systems. There are reliable rankings regularly updated by international agencies addressing such issues from different perspectives.

For instance:

https://worldpopulationreview.com/country-rankings/countries-with-freedom-of-speech

Of course, there is no statistical tool to handle data belonging to conceptual categories that do not exist (how one could provide a reliable estimate on unicorns?). Of course, there are no unicorns besides those from creative books (https://www.amazon.com/Natural-History-Unicorns-Chris-Lavers/dp/0060874147/ref=sr_1_1?crid=3J4POZ347FMN0&keywords=the+natural+history+of+unicorns&qid=1652124098&s=books&sprefix=the+natural+history+of+unicorns%2Cstripbooks-intl-ship%2C216&sr=1-1), whereas key populations do exist everywhere.

Key populations are NOT unicorns, but “unicorns” are a valuable “commodity” for a non-democratic society.

There is no way to extract data from governments who say that in the society “S” gay and other men who make sex with men do not exist cause this would an “abomination” or a” violation of sacred laws” (verbatim)… or that people who use substances deserve capital punishment and that they may exist here and there “as a failure of systems aiming to eliminate this ‘scourge’” (verbatim; I´m just using the words made public by different governments).

There are very interesting analyses about the pronounced influence of funding on the reliability and transparence of clinical science.

See, for instance:

https://pubmed.ncbi.nlm.nih.gov/19596837/

https://pubmed.ncbi.nlm.nih.gov/19596838/

https://pubmed.ncbi.nlm.nih.gov/18616022/

https://pubmed.ncbi.nlm.nih.gov/29401492/

https://pubmed.ncbi.nlm.nih.gov/31497290/

From my point of view, agencies and researchers should apply a similar reasoning to country-based information.

There is no doubt industry and partisan lobbyists may bias scientific findings (see, for instance: https://www.amazon.com/Merchants-Doubt-Handful-Scientists-Obscured/dp/1608193942/ref=sr_1_1?crid=1R8LNEJDL7T1A&keywords=merchants+of+doubt&qid=1652124771&s=books&sprefix=merchant%2Cstripbooks-intl-ship%2C233&sr=1-1)

Non-democratic societies should not be spared of critical scrutiny. The Discussion must be candid about it. NOT in the sense of criticizing A or B, but putting sensitive data on brackets.

Transparent information about the absence of free access to data could be easily shared with the potential readers using international rankings (see, for instance, the excellent reports by the UN agencies such as those available at: https://www.ohchr.org/en/instruments-and-mechanisms). The authors do not need to express their own points of view. The simple dissemination of available information would be very helpful.

Of course, is not our duty (or possible action) “to mend the world”, but to be critical about the information we disseminate. Unfortunately, we live in dark times (there is an ongoing war, but it is not called a “war”!).

There are several lessons to get from Hannah Arendt (e.g. https://www.amazon.com/Men-Dark-Times-Hannah-Arendt/dp/0156588900/ref=sr_1_1?crid=ZU0IKCW5ZQBW&keywords=men+in+dark+times&qid=1652275547&s=books&sprefix=men+in+dark+times%2Cstripbooks-intl-ship%2C214&sr=1-1).

They are not pleasant, but they say the truth, the plain truth.

Dear Chico, thank you for your thorough review and insights. While we agree with many of your comments, the purpose of our paper is not to make a political point but we do try and make a scientific point. We are calling out countries not producing reliable or any population sizes. We have added to our table of size estimations those countries that have used inadequate methods. For those countries that have no data, we have used imputation. We know this will upset some countries when they see their population sizes reported here, but hopefully this will prompt them to come up with their own, more reliable size estimations. We are also doing this to prove that countries that deny the existence of men who have sex with men or female sex workers, actually do have them. There are many countries listed here that may not like what they see. 

I have reviewed all of the links you provided and found many of them useful. However, for the citations on selective presentation of data, I see that most of these are from clinical surveys. Although this might be relevant for our paper, I think we do a good job at citing the countries that are providing questionable data in the table of population size estimation results. One of the papers suggests disclosing interests which may bias the reporting of data. We have done this by presenting our funding sources and affiliations. We are at a loss on how to incorporate unicorns into our paper and we have to admit that this paper would likely not meet up with the standards of Hannah Arendt. 

We have in the paper “Some estimates may be biased by poor implementation of methods, levels of stigma and discrimination in different contexts, and government pressure to minimize the mere existence of these populations.” We changed this sentence to “Some estimates are likely to be biased by poor implementation of methods, levels of stigma and discrimination in different contexts, and government pressure to minimize the mere existence of these populations.” We also point out how few countries are presenting any data and have added that this paper is, rather than waiting for estimates, are presenting them here with the hopes that more countries produce valid size estimations. 

6. PLOS authors have the option to publish the peer review history of their article (what does this mean?). If published, this will include your full peer review and any attached files.

Do you want your identity to be public for this peer review? For information about this choice, including consent withdrawal, please see our Privacy Policy.

Reviewer #1: No

Reviewer #2: Yes: Francisco Inacio Bastos

---

## [Editor Report · Decision Letter 1]

30 May 2022

Deriving and interpreting population size estimates for adolescent and young key populations at higher risk of HIV transmission: men who have sex with men and females who sell sex

PONE-D-21-18575R1

Dear Dr. Johnston,

We’re pleased to inform you that your manuscript has been judged scientifically suitable for publication and will be formally accepted for publication once it meets all outstanding technical requirements.

Kind regards,

Andrew R. Dalby, PhD

Academic Editor

PLOS ONE
---

## [Editor Report · Acceptance letter]

5 Aug 2022

PONE-D-21-18575R1 

Deriving and interpreting population size estimates for adolescent and young key populations at higher risk of HIV transmission: men who have sex with men and females who sell sex 

Dear Dr. Johnston:

I'm pleased to inform you that your manuscript has been deemed suitable for publication in PLOS ONE. Congratulations! Your manuscript is now with our production department. 

Kind regards, 

on behalf of

Dr. Andrew R. Dalby 

Academic Editor

PLOS ONE